



# Evaluating the impact of blowing snow sea salt aerosol on springtime BrO and O₃ in the Arctic

Jiayue Huang[1], Lyatt Jaeglé[1], Qianjie Chen[1,a], Becky Alexander[1], Tomás Sherwen[2], Mat J. Evans[2,3], Nicolas Theys[4], and Sungyeon Choi[5,6]

[1]Department of Atmospheric Sciences, University of Washington, Seattle, WA 98115, USA
[2]Wolfson Atmospheric Chemistry Laboratories, Department of Chemistry, University of York, York YO10 5DD, UK
[3]National Center for Atmospheric Science, University of York, York, YO10 5DD, UK
[4]Royal Belgian Institute for Space Aeronomy (BIRA-IASB), Brussels, Belgium
[5]Science Systems and Applications, Inc., Lanham, MD 20706, USA
[6]NASA Goddard Space Flight Center, Greenbelt, MD 20771, USA
[a]Now at Department of Chemistry, University of Michigan, Ann Arbor, MI 48109, USA

*Correspondence to*: Lyatt Jaeglé (jaegle@uw.edu)

**Abstract.** We use the GEOS-Chem chemical transport model to examine the influence of bromine release from blowing snow sea salt aerosol (SSA) on springtime bromine activation and O₃ depletion events (ODEs) in the Arctic lower troposphere. We evaluate our simulation against observations of tropospheric BrO vertical column densities (VCD$_{tropo}$) from the GOME-2 and OMI spaceborne instruments for three years (2007-2009), as well as against surface observations of O₃. We conduct a simulation with blowing snow SSA emissions from first-year sea ice (FYI, with a surface snow salinity of 0.1 psu) and multi-year sea ice (MYI, with a surface snow salinity of 0.05 psu), assuming a factor of 5 bromide enrichment of surface snow relative to seawater. This simulation captures the magnitude of observed March-April GOME-2 and OMI VCD$_{tropo}$ to within 17%, as well as their spatiotemporal variability (r=0.76-0.85). Many of the large-scale bromine explosions are successfully reproduced, with the exception of events in May, which are absent or systematically underpredicted in the model. If we assume a lower salinity on MYI (0.01 psu) some of the bromine explosions events observed over MYI are not captured, suggesting that blowing snow over MYI is an important source of bromine activation. We find that the modeled atmospheric deposition onto snow-covered sea ice becomes highly enriched in bromide, increasing from enrichment factors of ~5 in September-February to 10-60 in May, consistent with freshly fallen snow composition observations. We propose that this progressive enrichment in deposition could enable blowing snow-induced halogen activation to propagate into May and might explain our late-spring underestimate in VCD$_{tropo}$. We estimate that atmospheric deposition of SSA could increase snow salinity by up to 0.04 psu between February and April, which could be an important source of salinity for surface snow on MYI as well as FYI covered by deep snowpack. Inclusion of halogen release from blowing snow SSA in our simulations decreases monthly mean Arctic surface O₃ by 4-8 ppbv (15-30%) in March and 8-14 ppbv (30-40%) in April. We reproduce a transport event of depleted O₃ Arctic air down to 40º N observed at many sub-Arctic surface sites in early April 2007. While our simulation captures a few ODEs observed at coastal Arctic surface sites, it underestimates the magnitude of other events and entirely misses some events. We suggest that inclusion of direct snowpack activation, which is a strong local source of Br radicals in the shallow



Arctic boundary layer, could help reconcile the success of our simulation at capturing satellite retrievals of VCD$_{tropo}$ with its difficulty in reproducing local ODEs.

## 1 Introduction

Ozone depletion events (ODEs) are often observed in the springtime Arctic boundary layer (Barrie et al., 1988; Bottenheim et al., 2009, 1986; Bottenheim and Chan, 2006; Halfacre et al., 2014; Koo et al., 2012; Oltmans et al., 2012; Oltmans and Komhyr,

1986). While ODEs occur episodically at coastal Arctic sites, lasting 1-3 days, they are more persistent and widespread over the frozen ocean (e.g. Simpson et al., 2007 and references therein). These ODEs have been linked to the release of significant levels of bromine radicals, in a phenomenon called bromine explosion (Abbatt et al., 2012; Simpson et al., 2007b). Both ground-based and satellite instruments have reported elevated BrO columns over the Arctic during March and April (Choi et al., 2012, 2018; Hönninger and Platt, 2002; Jones et al., 2009; Liao et al., 2012; Neuman et al., 2010; Peterson et al., 2017;

Salawitch et al., 2010; Simpson et al., 2017; Theys et al., 2011; Wagner et al., 2001). In addition to ODEs in surface air during spring, the elevated bromine radicals result in the rapid oxidation of mercury and its deposition to snowpack (Ebinghaus et al., 2002; Schroeder et al., 1998; Steffen et al., 2008), which can have significant impacts on the health of people and wildlife in polar regions (AMAP, 2011).

An autocatalytic cycle involving the heterogeneous release of sea salt bromide (Br$^-$) via uptake of HOBr was suggested to be

the primary cause of bromine explosions and ODEs (Abbatt et al., 2012; Fan and Jacob, 1992; Hausmann and Platt, 1994; Simpson et al., 2007b):

$$HOBr + Br_{aq}^- + H_{aq}^+ \longrightarrow H_2O + Br_2 \tag{R1}$$

$$Br_2 + h\nu \longrightarrow 2Br \tag{R2}$$

$$Br + O_3 \longrightarrow BrO + O_2 \tag{R3}$$


$$BrO + HO_2 \longrightarrow HOBr + O_2 \tag{R4}$$

$$Br + CH_2O \rightarrow HBr \tag{R5}$$

$$HBr \xrightarrow{aerosol} Br_{aq}^- + H_{aq}^+ \tag{R6}$$

Reaction (R1) is a multiphase reaction, which takes place on acidic aerosols, cloud droplets, and snow grains. This reaction is required for the initial release of bromine to the atmosphere, but also maintains high levels of BrO by allowing fast recycling

of Br$^-$ to reactive bromine (Fan and Jacob, 1992; Lehrer et al., 2004). In addition, the heterogeneous reaction of HOBr with sea salt chloride (Cl$^-$) can release BrCl from the condensed phase (Abbatt et al., 2012).



In the global troposphere, inorganic bromine ($Br_y$) has three major sources: debromination of sea salt aerosol (SSA) produced by breaking waves in the open ocean, photolysis and oxidation of bromocarbons, and transport of $Br_y$ from the stratosphere. Release of $Br^-$ from oceanic SSA is estimated to be the largest global source of tropospheric $Br_y$ (Sander et al., 2003; Zhu et al., 2019). Current global models that include these three sources of tropospheric bromine as well as multiphase reactions such as (R1), cannot reproduce the observed elevated levels of tropospheric BrO over polar regions during spring (Parrella et al., 2012; Schmidt et al., 2016). Three classes of substrates specific to polar regions have been proposed as a source of $Br^-$ in reaction (R1): salty snowpack on sea ice and coastal regions (McConnell et al., 1992; Simpson et al., 2005; Toyota et al., 2011), first-year sea ice (Frieß et al., 2004; Nghiem et al., 2012; Wagner et al., 2007), SSA produced from frost flowers (Kaleschke et al., 2004; Rankin et al., 2002), and SSA produced from blowing salty snow (Jones et al., 2009; Yang et al., 2008).

Both laboratory and outdoor chamber experiments have detected $Br_2$ production when acidified surface saline snow was exposed to sunlight (Pratt et al., 2013; Wren et al., 2013). However, no $Br_2$ production was detected over sea ice and brine icicles, suggesting that brines and frost flowers on new sea ice surfaces are not a bromine activation source (Pratt et al., 2013). The proposed role of frost flowers as a direct source of SSA is disputed by several field and laboratory experiments showing that frost flowers are difficult to break even under strong wind conditions (e.g., Alvarez-Aviles et al., 2008; Roscoe et al., 2011; Yang et al., 2017).

A number of studies have focused on the role of aerosols, in particular SSA from blowing snow, as a key mechanism to initiate and sustain bromine activation aloft. Yang et al. (2008) proposed blowing snow SSA, produced after sublimation of wind-lofted salty snow on sea ice, as a source of BrO. This source is consistent with satellite observations of large-scale BrO column enhancements over Arctic and Antarctic sea ice, which are often associated with high surface wind speeds that can generate blowing snow events (Begoin et al., 2010; Blechschmidt et al., 2016; Choi et al., 2018; Jones et al., 2009). Using ground-based measurements in coastal Alaska, Frieß et al. (2004) found that periods of enhanced BrO were coincident with an increase in aerosol extinction at higher wind speeds (> 5 m s$^{-1}$), suggesting that halogen activation and/or recycling takes place in situ on SSA aerosol produced by sublimation of dispersed snow grains or frost flowers. Peterson et al. (2017) and Simpson et al. (2017) reported aircraft observations of the vertical distribution of BrO and aerosol extinction consistent with initial activation on snowpack followed by transport aloft (500-1000 m), where high BrO was sustained by recycling on aerosols. More recently, Frey et al. (2019) and Giordano et al. (2018) reported direct observations of SSA production from blowing snow above sea ice. In particular, Frey et al. (2019) found that the $Br^-/Na^+$ ratio of blowing snow SSA observed at 29 m above the ground decreased by a factor of 2-3 relative to observations at 2 m, suggesting rapid $Br^-$ release via (R1) from blowing snow SSA.

There are only a few modeling studies which have examined the link between blowing snow SSA and bromine explosions. In their pioneering study, Yang et al. (2010) implemented the Yang et al. (2008) blowing snow parameterization in the p-TOMCAT model and presented a comparison to satellite retrievals of BrO columns from the Global Ozone Monitoring



Experiment (GOME) for 2 months. Their evaluation showed qualitative agreement with observations over polar regions for
those months. Theys et al. (2011) describe two 3-day case studies over the Arctic and Antarctic comparing the p-TOMCAT
simulation to tropospheric BrO columns from the second Global Ozone Monitoring Experiment (GOME-2). There was
qualitative agreement with the location and timing of the two BrO explosion events. However, Theys et al. (2011) note that
beyond these case studies there were large discrepancies between model and observations, reflecting uncertainties in the p-
TOMCAT $Br^-$ emissions, such as assumptions about snow salinity and fraction of $Br^-$ in SSA released to the gas phase. Zhao
et al. (2016) used the UM-UKCA chemistry-climate model to simulate a bromine explosion case study initiated by a blowing
snow event and transported to the Canadian high Arctic. While the model reproduced the vertical extent of the BrO plume, it
was not as successful at capturing the timing or magnitude of the observed BrO. Choi et al. (2018) demonstrated a strong
spatial and temporal correlation between bromine explosions observed by OMI and blowing snow SSA as simulated in the
GEOS-5 modeling system. However, their study did not include a simulation of bromine photochemistry.

Here, we use the GEOS-Chem global chemical transport model to further quantify the role played by blowing snow SSA in
springtime Arctic bromine explosions and examine the associated tropospheric $O_3$ depletion. We systematically evaluate a 3-
year GEOS-Chem simulation (2007-2009) via comparisons to satellite retrievals of tropospheric BrO vertical column densities
($VCD_{tropo}$) from GOME-2 and the Ozone Monitoring Instrument (OMI). We also compare the model to in situ surface
observations of $O_3$ at several Arctic and sub-Arctic sites. Section 2 describes the GEOS-Chem simulations and the observations
used in this study. We evaluate the model's capability to reproduce the timing, magnitude, and spatial extent of bromine
explosion events observed by GOME-2 and OMI in Section 3. In section 4, we examine the impact of blowing snow SSA on
surface $O_3$. We assess the contribution of atmospheric deposition to surface snow salinity and bromide content in Section 5.
Conclusions are presented in Section 6.

## 2 Observations and model simulations

### 2.1 Satellite observations of tropospheric BrO vertical column densities

The Second Global Ozone Monitoring Experiment (GOME-2) is a nadir-scanning UV/visible spectrometer on the METOP-A
satellite, which was launched on 19 October 2006 in a sun-synchronous polar orbit with an equator crossing time of 9:30 LT
(Munro et al., 2006). The GOME-2 spectrometer covers the 240-790 nm wavelength region, with a spectral resolution between
0.26 and 0.51 nm. It has a ground pixel size of 80 km × 40 km and a scanning swath of 1920 km. In this study, we use the
daily 2007-2009 GOME-2 $VCD_{tropo}$ retrieved by Theys et al. (2011). Briefly, the $VCD_{tropo}$ are derived from a residual technique
that combines measured slant columns and calculated stratospheric columns, accounting for the impact of clouds, surface
reflectivity and viewing geometry on the measurement sensitivity. The stratospheric BrO contribution is removed using the
dynamic climatology based on the BASCOE CTM as described in Theys et al. (2009, 2011). The tropospheric slant columns
are converted to vertical columns with a tropospheric air mass factor (AMF) assuming different BrO profile shapes depending
on surface albedo. For low surface albedo (<0.5), a Gaussian shape BrO profile with a maximum at 6 km is used. For high





surface albedo (>0.5), the assumed tropospheric BrO concentration profile is constant in the first km above the Lambertian surface reflector. For this second case, which characterizes most of the ice- and snow-covered Arctic, there is a high sensitivity to BrO close to the surface and a weak dependence of the tropospheric AMF to the shape of the profile. As in Theys et al. (2011), we only consider retrievals for solar zenith angles (SZA) less than 80° and cloud fractions below 0.4 for which the pressure difference between the surface and the top of the cloud less than 400 mbar.

We also use the BrO $VCD_{tropo}$ from the Ozone Monitoring Instrument (OMI) onboard the Aura satellite, which was launched on 15 July 2004 in a sun-synchronous polar orbit with a 13:30 LT equator overpass time (Levelt et al., 2006). OMI is a nadir solar backscatter spectrometer that measures ultraviolet–visible wavelengths (270–500 nm), and has a horizontal resolution of 13 km × 14 km and swath width of 2600 km. Since 2008, the OMI swath coverage has been reduced due to an external obstruction. We use here the daily 2007-2009 $VCD_{tropo}$, which were retrieved by Choi et al. (2018) for the months of March and April poleward of 60°N. The total BrO slant column densities are derived from the OMI total BrO (OMBRO) product, by fitting a model function to OMI UV backscattered radiance at 319–347.5 nm (Choi et al., 2018). The $VCD_{tropo}$ are retrieved using the same residual technique and dynamic climatology stratospheric BrO columns as in Theys et al. (2011). In the OMI tropospheric AMF calculation, the assumed BrO shape profile is based on a composite of aircraft measurements obtained during the NASA Arctic Research of the Composition of the Troposphere from Aircraft and Satellites (ARCTAS) campaign (Choi et al., 2012). We use the same selection criteria as in Choi et al. (2018): SZA<80°, surface reflectivity>0.6, and only retain observations with low cloud contamination (difference between OMI rotational Raman cloud pressure and terrain pressure <100 hPa). The uncertainty of OMI and GOME-2 $VCD_{tropo}$ retrieved over the highly reflective surfaces of polar regions is on the order of 30-50%.

## 2.2 Surface $O_3$ measurements

We use hourly in situ $O_3$ measurements at three Arctic surface sites: Utqiaġvik, Alaska (also known as Barrow, 71.3º N, 156.6º W, 8 m above sea level) from the NOAA Earth System Research Laboratory, Alert, Nunavut (82.5º N, 62.5º W, 187 m) from the Canadian Air and Precipitation Monitoring Network (CAPMoN), and Zeppelin, Spitzbergen (78.9º N, 11.8º E, 474 m) from the Norwegian Institute for Air Research. We also use hourly surface $O_3$ measurements at several sub-arctic sites from CAPMoN as well as from the Canadian National Air Pollution Surveillance (NAPS) network and from the United States Clean Air Status and Trends Network (CASTNET). For the NAPS sites, we only consider sites sampling background air in categories "Forest" and "Undeveloped Rural", and for CASTNET we exclude sites located in "Urban/Agricultural" areas. This selection allows us to avoid more polluted sites, which can be subject to low winter-spring $O_3$ concentrations associated with NO titration of $O_3$. The subarctic sites include Bonner Lake (49.4º N, 82.1º W, 242 m), Algoma (47.0º N, 84.38º W, 411 m), and Egbert (44.2º N, 79.8º W, 206 m) from the CAPMoN network, as well as Elk Island (53.68º N, 112.87º W, 714 m) from NAPS and Woodstock (43.94º N, 71.70º N, 255 m) from CASTNET.



## 2.3 The GEOS-Chem chemical transport model

GEOS-Chem is a 3-D global chemical transport model (Bey et al., 2001). In this work, we use GEOS-Chem v11-02d (http://www.geos-chem.org, last access 19 August 2019), driven by the Modern-Era Retrospective analysis for Research and

Applications, version 2 (MERRA-2) assimilated meteorological fields (Gelaro et al., 2017), which have a native horizontal resolution of 0.5º latitude by 0.625º longitude with 72 vertical levels. We regrid the MERRA-2 fields to a 2º× 2.5º horizontal resolution and 47 vertical levels with merged levels above 80 hPa for computational expediency. For the time-period of the simulations conducted here (2007-2009), the daily boundary conditions for sea ice concentrations in MERRA-2 are from the high-resolution (1/20º) Operational SST and Sea Ice Analysis (OSTIA), which uses daily sea ice concentration products from

multiple Special Sensor Microwave Imager (SSM/I) satellites (Donlon et al., 2012).

Global anthropogenic emissions are from EDGAR v4.2 (Olivier and Berdowski, 2001) for 1970-2008. Biomass burning emissions are from the Global Fire Emissions Database version 4 (GFEDv4) emission inventory (van der Werf et al., 2017). Biogenic emissions of volatile organic compounds (VOCs) are from the Model of Emissions of Gases and Aerosols from Nature version 2.1 (MEGAN 2.1) (Guenther et al., 2012).

GEOS-Chem simulates detailed $HO_x$-$NO_x$-VOC-$O_3$-halogen-aerosol tropospheric chemistry. The chemical mechanism in GEOS-Chem v11-2d was updated to the most recent JPL/IUPAC recommendations, as described in (Fischer et al., 2014; Fisher et al., 2016; Mao et al., 2013; Travis et al., 2016). This version of GEOS-Chem includes bromine-chlorine-iodine halogen chemistry. The bromine chemistry mechanism was first described in Parrella et al. (2012). The mechanism was then updated by Schmidt et al. (2016) to include extensive multiphase chemistry as well as coupling to tropospheric chlorine

chemistry, which provides an important pathway to recycle bromine radicals. The uptake coefficients for the heterogeneous reactions of HOBr, $ClNO_3$ and $O_3$ with $Br^-$ in aerosols, as well as for the reaction of HOBr with $Cl^-$ in aerosols follow Ammann et al. (2013). The uptake coefficient between $O_3$ and $Br^-$ considers both bulk and surface reactions. Sherwen et al. (2016a, 2016b) implemented iodine chemistry and Cl-Br-I interactions. Chen et al. (2017) added the in-cloud oxidation of dissolved $SO_2$ (S(IV)) by HOBr in GEOS-Chem, which decreased the global $Br_y$ burden by 50% and resulted in improved agreement

with GOME-2 $VCD_{tropo}$ between 60º N and 60º S. Fifteen bromine tracers are transported ($Br_2$, Br, BrO, HOBr, HBr, $BrNO_2$, BrCl, $BrONO_2$, $CHBr_3$, $CH_2Br_2$, $CH_3Br$, IBr, $CH_2IBr$, $Br^-$ on accumulation mode SSA and $Br^-$ on coarse mode SSA), with sources from photolysis of $CHBr_3$, oxidation of $CHBr_3$, $CH_2Br_2$ and $CH_3Br$ by OH radicals, transport of reactive bromine from the stratosphere, and SSA debromination driven by explicit heterogeneous reactions of SSA $Br^-$ with HOBr, $ClNO_3$, and $O_3$.

Open ocean emissions of SSA are a function of wind speed and sea surface temperature (SST) as described in Jaeglé et

al.(2011), with updates from Huang and Jaeglé (2017) for cold ocean waters (SST < 5°C). Two separate SSA tracers are transported: accumulation mode SSA ($r_{dry}$ = 0.01−0.5 μm) and coarse mode SSA ($r_{dry}$ = 0.5–8 μm). Sea salt $Br^-$ is emitted



assuming a ratio of $2.11 \times 10^{-3}$ kg Br per kg of dry SSA, based on the composition of sea water (Lewis and Schwartz, 2004; Sander et al., 2003). Sea salt Br⁻ is transported in two tracers as part of accumulation and coarse mode SSA.

### 2.3.1 The blowing snow SSA simulation in GEOS-Chem

The blowing snow SSA simulation in GEOS-Chem is described in Huang and Jaeglé (2017) and Huang et al. (2018). Blowing snow SSA emissions are a function of relative humidity, temperature, age of snow, surface snow salinity and wind speed following the parameterization of Yang et al. (2008, 2010). As in Huang and Jaeglé (2017), we assume N=5 for the number of SSA particles produced per snowflake, and a mean snow age of 3 days over the Arctic. Two key parameters controlling the magnitude of blowing snow SSA production and subsequent heterogeneous Br⁻ release are the surface snow salinity and Br⁻

content, both of which have very few observational constraints. Sea salt and Br⁻ in surface snow originates from upward migration of brine from sea ice, atmospheric deposition of SSA as well as gas-phase reactive bromine species, and contamination of snow by frost flowers (Abbatt et al., 2012). Domine et al. (2004) estimates that frost flowers could account for 10% of the observed surface snow salinity. Upward migration of brine is expected to be the dominant source of salinity for thin snow over sea ice (<10-17 cm) (Domine et al., 2004; Peterson et al., 2019). As the snow depth gets thicker, the salinity of

surface snow decreases and the influence of atmospheric deposition likely becomes more important (Krnavek et al., 2012; Nandan et al., 2017). Snow over first-year sea ice (FYI) is typically more saline than over multi-year sea ice (MYI), as MYI is desalinated by flushing and gravity drainage during repeated summer melt cycles.

Krnavek et al. (2012) sampled the chemical composition of surface snow on land-fast sea ice near Utqiaġvik, Alaska. They reported a median salinity of 0.7 practical salinity unit (psu) for 2-3 weeks old FYI, 0.1 psu for thicker FYI, and 0.01 psu for

MYI. Domine et al. (2004) reported a salinity of 0.02 psu for MYI near Alert. Peterson et al. (2019) measured surface snow in FYI and MYI regions near Greenland, Alaska, as well as in the central Arctic, finding higher mean salinities for snowpacks less than 17 cm deep (0.15 psu) compared to deeper snowpacks (0.02 psu). The observed Br⁻ concentrations in surface snow over Arctic sea ice is highly variable, ranging from $10^{-2}$ to $10^{3}$ μM (Domine et al., 2004; Krnavek et al., 2012; Peterson et al., 2019). This high variability is accompanied with either depletion or enhancement in the bromide to sodium (Br⁻/Na⁺) ratio

relative to sea water composition. Depletion in Br⁻ relative to seawater indicates loss to the atmosphere via heterogeneous reactions. Enhancements can be the result of precipitation of hydrohalite (NaCl-2H₂O) below 251K in brine (Koop et al., 2000; Morin et al., 2008). Indeed, the laboratory measurements of aqueous NaCl and sea-salt solution droplets of Koop et al. (2000) show a factor of 12 increase in the Br⁻/Na⁺ ratio at 240K relative to 273K. Enhancements in Br⁻ could also be caused by the deposition of atmospheric aerosol and HBr produced during bromine explosions onto surface snow (Simpson et al., 2005,

2007b). Domine et al. (2004) report measurements of surface snow on sea ice near Alert, with a factor of 5 enrichment in Br⁻ relative to seawater, while surface snow over an Arctic Ocean site further north displayed a Br⁻ enrichment of 25. Samples of surface snow on MYI appear to be enhanced in Br⁻ more often than on FYI (Krnavek et al., 2012), which suggests that MYI, in addition to FYI, could play an active role in Arctic boundary layer bromine and chlorine chemistry (Peterson et al., 2019).



We conduct three GEOS-Chem simulations as part of this work: a standard simulation (referred to as "STD") in which the

only source of SSA is from the open ocean, and two simulations in which we add a blowing snow source of SSA. In the first blowing snow simulation (referred to as "FYI Snow"), we assume an Arctic surface snow salinity of 0.1 psu on FYI and 0.01 psu for MYI. In the second blowing snow simulation ("FYI+MYI Snow"), we use a salinity of 0.1 psu on FYI and 0.05 psu on MYI. These two salinities assumed for MYI are used to examine the role of bromine activation on MYI. For both blowing snow simulations, we assume a surface snow $Br^-$ enrichment factor of 5 relative to sea water (sea salt $Br^-$ is emitted as assuming

a ratio of $10.55 \times 10^{-3}$ kg Br per kg of dry blowing snow SSA emitted). As sea ice age is not tracked in MERRA-2, for each year we identify the location of Arctic MYI from the preceding September minimum sea ice extent in MERRA-2. The FYI extent is calculated by subtracting the MYI extent from the total sea ice extent. Note that this very simple approach does not account for advection and melting of MYI between one September to the next. All simulations are initialized with a 6-month spin-up in 2006 and then followed by a 3-year simulation for 2007-2009.

In previous work (Huang et al., 2018; Huang and Jaeglé, 2017), we found that the GEOS-Chem blowing snow simulation (using 0.1 psu on FYI and 0.01-0.1 psu on MYI) reproduced the seasonal cycle and magnitude of SSA mass concentrations observed at Utqiaġvik, Alert, and Zeppelin, but that the STD simulation underestimates observations by a factor of 2-10 during winter and spring. The blowing snow simulations also reproduced the seasonal cycle of aerosol extinction coefficients observed in the lower troposphere (0-2 km) over Arctic sea ice by the Cloud-Aerosol Lidar with Orthogonal Polarization

(CALIOP) on-board the CALIPSO satellite (Huang et al., 2018).

For comparison with satellite retrievals of VCD$_{tropo}$, we sample the model at 9:00-11:00 LT, which corresponds to the overpass times of GOME-2 in the Northern Hemisphere. We found that sampling GEOS-Chem at 13:00-15:00 LT, matching the OMI overpass time, results in less than a 1% difference in VCD$_{tropo}$. We thus only show the model results for 9:00-11:00 LT. We regrid daily OMI and GOME-2 VCD$_{tropo}$ to a horizontal resolution of $2° \times 2.5°$ for comparison to GEOS-Chem. The model is

sampled on days and locations of GOME-2 and OMI overpasses with available retrievals.

## 3 Evaluation of the impact of blowing snow SSA on BrO VCD$_{tropo}$

### 3.1 Seasonal cycle of VCD$_{tropo}$ in the Northern Hemisphere

Several previous GEOS-Chem model versions have been evaluated against GOME-2 BrO VCD$_{tropo}$. Schmidt et al. (2016) showed that the inclusion of SSA debromination resulted in a 50-100% overestimate in BrO in the northern hemisphere.

Sherwen et al. (2016a, 2016b) added iodine chemistry and disabled SSA debromination, finding relatively good agreement with GOME-2 observations over the Arctic and the summertime low- and mid- latitudes. Chen et al. (2017) enabled SSA debromination and added in-cloud oxidation of dissolved S(IV) with HOBr to the model version of Schmidt et al. (2016) without iodine chemistry. They found improved agreement with GOME-2 VCD$_{tropo}$ over low- and mid- latitudes, but factors





of 3-10 underestimate over high latitudes. Our STD simulation includes SSA debromination, in-cloud oxidation of dissolved
S(IV) by HOBr, as well as iodine chemistry, and chlorine chemistry.

Fig. 1 (panels a and b) shows that with these four components, the STD simulation agrees well with the GOME-2 $VCD_{tropo}$ at
0-30° N and 30-60° N averaged for 2007-2009. Over the Arctic (>60° N), however, the STD simulation underestimates
GOME-2 and OMI observations by up to 50% during spring (Fig. 1c). Note that we do not show monthly mean GOME-2
$VCD_{tropo}$ at 60-90° N for November-February, as less than 70% of the polar region has valid data (SZA is generally greater
80° for these months).

The 2007-2009 GOME-2 and OMI $VCD_{tropo}$ display a March-April maximum of ~3-3.5×$10^{13}$ cm$^{-2}$ poleward of 60° N (Fig.
1c). The inclusion of blowing snow SSA emissions in the GEOS-Chem FYI Snow simulation increases the modeled springtime
$VCD_{tropo}$ by 42% in March and 52% in April, improving the agreement with satellite retrievals (Fig. 1c). The FYI+MYI Snow
simulation increases the modeled $VCD_{tropo}$ by another 10-20% in March and April. Both blowing snow simulations are within
5-20% of observed $VCD_{tropo}$ in March and April, however they predict too rapid a decrease in $VCD_{tropo}$ in May.

In late summer and fall, when sea ice extent is at its minimum and blowing snow SSA emissions are negligible, GEOS-Chem
underestimates GOME-2 $VCD_{tropo}$ by 30-40%. Over the cloudy summer Arctic, the reaction HOBr+S(IV) in cloud water
provides a sink for HOBr and decreases the $Br_y$ abundance by about 70−90% (Chen et al., 2017). We hypothesize that including
acid displacement HCl from SSA would lead to an increase in HOBr because of the competition between HOBr+Cl$^-$ (which
recycles HOBr by producing BrCl) and HOBr+S(IV) (which is a sink for HOBr) in cloud droplets. Indeed, in a subsequent
version of GEOS-Chem, Wang et al. (2018) added this source of HCl, finding an increase in BrO, especially in cloudy high
latitudes.

### 3.1 Spatial distribution of springtime Arctic BrO $VCD_{tropo}$

Fig. 2 shows the spatial distribution of monthly mean $VCD_{tropo}$ during March and April 2007-2009. In March, both GOME-2
and OMI exhibit enhanced $VCD_{tropo}$ (3.5-5×$10^{13}$ cm$^{-2}$) in an arc between Baffin Bay - along the west coast of Greenland - and
the Laptev Sea, off the Northern coast of Siberia. In April, the observed $VCD_{tropo}$ reach values >3.5-4×$10^{13}$ cm$^{-2}$ over most of
the sea ice covered Arctic Ocean. Fig. 3a compares the GOME-2 and OMI $VCD_{tropo}$ poleward of 60°N for March and April
2007-2009, showing that they are highly correlated (r=0.92, slope=0.99), with OMI being 13% higher than GOME-2.

As shown in Figs. 2 and 3d, the GEOS-Chem STD simulation predicts $VCD_{tropo}$ lower than 2.5-3×$10^{13}$ cm$^{-2}$ in March-April,
with little spatial variability compared to observations (r=0.22-0.36). The normalized mean bias (NMB = 100 × $\sum$(M$_i$-O$_i$)/$\sum$O$_i$,
with observations O$_i$ and model M$_i$ summed poleward of 60° N) is -41% compared to OMI and -34% compared to GOME-2.
The FYI Snow simulation has more success at reproducing the magnitude of the observed $VCD_{tropo}$ in March and April
(NMB=-12% relative to OMI and NMB=+1% relative to GOME-2), as well as their spatial distribution (r=0.70-0.76). The


$VCD_{tropo}$ predicted by the FYI+MYI Snow simulation agrees better with OMI (NMB=+2% and r=0.76), but overestimates
GOME-2 (NMB=+17% and r=0.79). Similar to the satellite retrievals in March, the FYI and FYI+MYI snow simulations
displays the largest $VCD_{tropo}$ over the Canadian Arctic Archipelago (Fig. 2), with secondary maxima over Baffin Bay, the
Greenland Sea, and the east Siberian Sea. In April, the $VCD_{tropo}$ in the FYI snow simulation underestimates GOME-2 and OMI
throughout the Arctic, but the FYI+MYI snow simulation overestimates satellite observations in the high Arctic (>80° N),
while underestimating observations at 70-80° N. This suggests spatial variability in salinity and/or Br$^-$ enhancement for snow
that is not captured in our assumptions of spatially uniform values for FYI and MYI. We also find that the simulated $VCD_{tropo}$
underestimate observations over the coastal Arctic, especially in April. This could be due halogen activation from snow over
land (Simpson et al., 2005), which is not considered in our simulation as we assume zero salinity and Br$^-$ content in continental
snow.

### 3.2 Daily variations in pan-Arctic BrO $VCD_{tropo}$

Fig. 4a shows that GOME-2 daily mean $VCD_{tropo}$ poleward of 60° N is highest in March-April, and then decreases in May.
The OMI $VCD_{tropo}$ in March and April displays a similar day-to-day variability as GOME-2. Overall the FYI+MYI Snow
simulation shows good agreement with the magnitude and daily variability of the combined mean GOME-2 and OMI $VCD_{tropo}$
(NMB=-8.3%, r=0.85). While the FYI+MYI Snow simulation reproduces the March-April $VCD_{tropo}$, it predicts too rapid a
decrease in May for all three years (Fig 4a). The difference between the FYI Snow and MYI+FYI Snow simulations is the
largest in April as polar sunrise reaches the high Arctic, where the majority of MYI is located. Most of the BrO enhancements
in the two blowing snow simulations occur below 1 km altitude, occasionally extending to 2 km (see Fig. S1ab).

As discussed in previous studies (Choi et al., 2018; Nghiem et al., 2012; Richter et al., 1998; Salawitch et al., 2010; Theys et
al., 2011) satellite observations show the frequent occurrence of "BrO hotspots" over Arctic sea ice during spring. For this
study, we define a BrO hotspot when local $VCD_{tropo}$ exceed $4.5 \times 10^{13}$ cm$^{-2}$, which is the 90$^{th}$ percentile of GOME-2 $VCD_{tropo}$
poleward of 60° N in March-April. For OMI, the corresponding 90$^{th}$ percentile is $5.1 \times 10^{13}$ cm$^{-2}$. We use these respective
criteria to calculate the daily area covered by BrO hotspots as observed by GOME-2 and OMI. The episodic nature of GOME-
2 and OMI BrO hotspots is apparent in Fig. 4b, which shows a large degree of daily variability, with events lasting between a
few days to 2 weeks. The green shading in Fig. 4 identifies multi-day events (>5 days) when the GOME-2 hotspot area exceeds
2 million km$^2$. There are 2 such events in both 2007 and 2008, and 3 events in 2009 (Fig. 4b). These events reach maximum
extents of 4-6 million km$^2$. For comparison, the mean sea-ice extent during March-April poleward of 60°N is ~10 million km$^2$
(with a ~65%/35% split between FYI and MYI).

We apply the OMI threshold ($VCD_{tropo} > 5.1 \times 10^{13}$ cm$^{-2}$) to the GEOS-Chem simulations. On average, the FYI+MYI Snow
simulation reproduces the observed spatial extent of BrO hotspots to within 4% and captures the timing of the 2 large-scale
episodes in 2007 (26 March – 4 May, 10–20 May 2007), the two episodes in 2008 (7–17 March and 1–8 April 2008). For



2009, the FYI+MYI Snow simulation predicts two episodes (5–15 March and 11–23 April), but the aerial extent of these episodes is overestimated, and the simulation misses the third episode on 6–19 May 2009 (Fig. 4b). In the model, the variability in BrO hotspots is driven by the temporal variations of the blowing snow SSA burden (Fig. 4c).

### 3.3 Two case studies of large BrO explosion events

Fig. 5 shows the daily spatial distribution of $VCD_{tropo}$ on 25–30 March 2007, corresponding the largest BrO hotspot episode
observed by GOME-2 and OMI during our study period (Fig. 4ab). The GOME-2 and OMI $VCD_{tropo}$ exceed $7\times10^{13}$ cm$^{-2}$ (up to $15\times10^{13}$ cm$^{-2}$) over the sea-ice covered region in the high Arctic region (>70°N). The enhanced $VCD_{tropo}$ start on 25 March over the East Siberian Sea and then rotate counterclockwise, reaching the Beaufort Sea on 27 March and the Canadian Archipelago on 28-30 March 2007. This event was previously discussed in Begoin et al. (2010) and Choi et al. (2018). Begoin et al. (2010) used the FLEXPART particle dispersion model to link the enhanced GOME-2 $VCD_{tropo}$ to a cyclone with very
high surface wind speeds, favorable for generating blowing snow SSA. Choi et al. (2018) found that the spatial pattern of OMI $VCD_{tropo}$ was consistent with that of the GEOS-5 simulated blowing snow SSA burden on both FYI and MYI, while restricting the blowing snow emissions to FYI only did not agree as well with the observed $VCD_{tropo}$. Our FYI+MYI Snow simulation reproduces the evolution of this BrO hotstpot, although the magnitude of the modeled $VCD_{tropo}$ overestimates observations after 28 March 2007. The main difference between the FYI Snow and FYI+MYI Snow simulations is on 25-26 March 2007,
when the high winds are located over the North Pole where MYI is (Figs. 5 and S2). As there are no valid GOME-2 and OMI retrievals at these high latitudes (SZA>80°), we cannot assess which simulation agrees better with observations for this case study.

Fig. 6 shows the daily distribution of $VCD_{tropo}$ on 14-19 April 2007. During this event, there is a factor of 1.5 difference in pan-Arctic $VCD_{tropo}$ calculated with the FYI Snow and FYI+MYI Snow simulations (Fig. 4a). The GOME-2 and OMI $VCD_{tropo}$
show high values (>$5\times10^{13}$ cm$^{-2}$) being maintained over the North Pole (Fig. 6). The MERRA-2 meteorological fields predict high surface winds in that area, which is covered by MYI (Fig. 7, bottom row). The $VCD_{tropo}$ calculated by the FYI Snow simulation show little enhancement over the North Pole (Fig. 7), while the FYI+MYI Snow simulation better captures the magnitude and shape of the observed enhancement (Fig. 6), suggesting that blowing snow SSA emissions over MYI can be a significant source of BrO.

Overall, we find that the FYI+MYI Snow simulation captures reasonably well the spatial and temporal distribution of the observed $VCD_{tropo}$, as well as the spatial extent and frequency of BrO hotspots in March and April. In our simulation, blowing snow SSA emissions on MYI account for 20-30% of the $VCD_{tropo}$ enhancement and could thus represent an important source of bromine activation. This is consistent with previous studies, which have suggested that, in addition to FYI, MYI could play a significant role in halogen activation. Gilman et al. (2010) used backtrajectories to calculate sea ice exposure of air masses
observed during the International Chemistry Experiment in the Arctic Lower Troposphere (ICEALOT) ship-based study,



finding that exposure to both FYI and MYI was the best predictor of reduced $O_3$ levels. Choi et al. (2018) reported that the frequency of springtime BrO explosion events observed by OMI for 2005-2015 was strongly correlated with blowing snow SSA emissions over all sea ice, but they found little to no correlation when only FYI was considered as a source of blowing snow SSA. More generally, low $O_3$ and/or high BrO $VCD_{tropo}$ have been observed over and downwind of regions with a high

fraction of MYI such as the Canadian Archipelago and the eastern Beaufort Sea (Bottenheim and Chan, 2006; Choi et al., 2012; Halfacre et al., 2014; Koo et al., 2012; Richter et al., 1998; Salawitch et al., 2010). In contrast to the generally good agreement in March and April, during May our simulation is systematically too low compared to GOME-2 $VCD_{tropo}$ (Figs. 1c and 4ab). In Section 5 we will discuss how the recycling of bromine deposition to snowpack in late spring could propagate blowing snow-induced halogen activation into May.

## 4 Impact of blowing snow SSA on tropospheric $O_3$ over the Arctic

### 4.1 Comparison to hourly surface $O_3$ observations

We evaluate the ability of our simulations to capture ODEs via comparisons to hourly $O_3$ observations at several Arctic and sub-Arctic sites in 1 March - 31 May 2007 (Fig. 8). Observations at Utqiaġvik and Alert show frequent occurrences of low $O_3$ mixing ratios (<10 ppbv), maintained for 1-7 days. These events are somewhat less frequent at Zeppelin. The FYI+MYI Snow

simulation reproduces some of these events (such as the $8 - 18$ April and 26 April $-$ 10 May depletion events at Utqiaġvik, the April events at Alert, many of the events at Zeppelin between late March to late April), but misses many others, in particular several March and late May events at Utqiaġvik as well as the late April-May events at Alert.

Observed surface $O_3$ depletion events can be caused by advection of upwind $O_3$ poor air and/or local depletion (Bottenheim and Chan, 2006; Halfacre et al., 2014; Hopper et al., 1998; Jacobi et al., 2006; Simpson et al., 2007b, 2017). The rapid recovery

of surface $O_3$ after a depletion event is often due to turbulent vertical mixing of $O_3$-rich free tropospheric air down to the surface or to advection of $O_3$-rich continental air (Bottenheim et al., 2009; Gong et al., 1997; Hopper et al., 1998; Jacobi et al., 2010; Moore et al., 2014; Morin et al., 2005). Furthermore, some ODEs can be highly localized. Using ice-tethered buoys over coastal regions and over the Arctic Ocean, Halfacre et al. (2014) found that large areas of the Arctic Ocean are partially depleted in $O_3$ during spring with local imbedded areas (~200-300 km) that are more depleted. Morin et al. (2005) report that

dynamic conditions of the boundary layer near the shore can lead to very different surface $O_3$ behaviors observed at Alert compared to at a site over the sea ice 10 km away. Similar results were found by Jacobi et al. (2006), when comparing $O_3$ observations at Zeppelin Station and nearby (~100-200 km) ship-based measurements. The coarse resolution of our simulation ($2°×2.5°$) and the general difficulty of models in reproducing boundary layer depth and vertical mixing processes could thus be one explanation for our poor representation of ODEs at some of these surface sites. Another explanation is the lack of

detailed chlorine chemistry in this version of GEOS-Chem. In particular, we do not consider acid displacement of $Cl^-$ in SSA which, together with bromine chemistry, could act to enhance ODEs. Finally, our simulation does not include local snowpack Br activation, which has been shown to lead to surface ODEs when the stable Arctic boundary layer is decoupled from





convective exchange with the free troposphere (Custard et al., 2017; Peterson et al., 2015, 2017; Pratt et al., 2013; Wang et al., 2019).

## 4.2 Spatiotemporal distribution of O₃ depletion in GEOS-Chem

Fig. 9 shows the spatial distribution of monthly mean surface $O_3$ mixing ratios calculated in the FYI+MYI Snow simulation as well as the decrease in surface $O_3$ ($\Delta O_3$) relative to the STD simulation for March and April 2007-2009. The lowest monthly mean $O_3$ mixing ratios are ~25 ppbv in March over the Canadian Archipelago and ~20-25 ppbv in April over the North Pole (Fig. 9a), with a spatial distribution corresponding to that of the simulated $VCD_{tropo}$ (Fig. 2). Note that the low $O_3$ values simulated over Europe are due to $NO_x$ titration. The monthly mean values from the FYI+MYI Snow simulation are within better than 8 ppbv of surface observations at Utqiaġvik (model/obs: March 28/22 ppbv, April 28/20 ppbv), Alert (model/obs: March 29/33 ppbv, April 27/27 ppbv), Zeppelin (model/obs: March 32/39 ppbv, April 27/31 ppbv), and Elk Island (model/obs: March 28/36 ppbv, April 35/38 ppbv). The mean values of surface $\Delta O_3$ increase from 4-8 ppbv (15-30% depletion relative to the STD simulation) in March, to 8-14 ppbv (30-40%) in April, when sufficient sunlight is available to drive photochemistry at high latitudes (Fig. 9b). Poleward of 60º N, the FYI+MYI Snow simulation displays a pan-Arctic $O_3$ decrease of 3.7 ppbv (11%) in March and 8.3 ppbv (23%) relative to the STD simulation.

We define the frequency of ODEs as the percent of time when more than 20 ppbv $O_3$ is lost due to blowing snow ($\Delta O_3 >$ 20 ppbv, Fig. 9c). Applying this definition to the FYI+MYI Snow simulation, we find that ODEs occur up to 1-5% of the time in March, increasing to up to 15-25% of the time in April as sunlight extends to higher latitudes (Fig. 9c). Using an ODE definition including more moderate events ($\Delta O_3 >$ 10 ppbv), we find that poleward of 70º N ODEs occur 20-60% of the time in April (Fig. 9d).

The decrease in surface $O_3$ due to blowing snow SSA is 60% larger in the FYI+MYI Snow simulation compared to the FYI Snow simulation (Fig. 10a). The timeseries of pan-Arctic daily mean $\Delta O_3$ for the FYI+MYI Snow simulation at different altitudes shows that $\Delta O_3$ is uniform in the bottom 500 m altitude and extends to 1000 m altitude, where the model-calculated $\Delta O_3$ is about half of the surface $\Delta O_3$. At 2000 m, $\Delta O_3$ is generally below 4 ppbv (Fig. 10b). This is consistent with ozonesonde profiles, which indicate that depletion events are confined to the lowest 1000 m with an average height of the top of the layer at 500 m (Hopper et al., 1998; Oltmans et al., 2012; Tarasick and Bottenheim, 2002). We find that the Arctic tropospheric $O_3$ burden (>60º N) decreases by 3% in March and 6% in April in the FYI+MYI simulation relative to the STD simulation.

In our simulation, the timing of the maximum $\Delta O_3$ takes place 4-5 days after the maximum in the blowing snow SSA burden (Figs. 4c and 10b), reflecting increasing $O_3$ loss as BrO concentrations increase. This is illustrated during the March 25-30 2007 event: the pan-Arctic blowing snow SSA burden and $VCD_{tropo}$ reach their maximum value on 27 March, while the lowest values for surface $O_3$ occur 4 days later on 31 March (Fig. 5, bottom row). By 29-30 March the blowing snow SSA burden is back to low values over the Arctic, but the $VCD_{tropo}$ are still elevated and low surface $O_3$ mixing ratios are predicted throughout


the Arctic. This can be explained by the different lifetimes of these species over the springtime Arctic: ~1 day for SSA, 4-7 days for the $Br_y$ family (consisting of all the gas-phase inorganic bromine species), and 30-40 days for $O_3$.

**4.3 Transport of $O_3$-depleted Arctic air to lower latitudes**

In some instances, large-scale blowing snow induced bromine activation can influence surface $O_3$ in sub-Arctic regions. The large BrO explosion and associated $O_3$ depletion from late March to early April 2007 (Figs. 4ab) illustrates this. The green shading in Fig. 8 corresponds to the times when this BrO explosion is transported to the three Arctic sites and several sub-

Arctic sites. The GEOS-Chem FYI+MYI Snow simulation shows transport of $O_3$-depleted air to Utqiaġvik on 25-27 March 2007, with observations showing much stronger depletion than the model (Fig. 8a). This air mass is observed at Alert on 31 March-4 April and the eastern edge of that $O_3$-depleted air reaches Zeppelin around the same time (Figs. 8bc). The $O_3$-depleted air is then transported southward over Hudson Bay and towards the northeastern United States during 4-9 April (Figs. 5 and 11), mostly below 1,000 m altitude. At the synoptic level, a surface low was sweeping across the United States from the central

Rockies to the Great Lakes in early April, bringing sub-freezing Arctic air behind its cold front with record-breaking cold temperatures being measured over the Central Plains and much of the Southeast (NOAA/USDA, 2008).

Between 2 April and 7 April, observed $O_3$ mixing ratios at 5 sub-Arctic sites from Elk Island (53.7° N) to Woodstock (43.9° N), decreased from background levels of 40 ppbv down to 10-20 ppbv (Fig. 8). This decrease is reproduced by both the FYI and FYI+MYI Snow simulations, which show that enhanced $VCD_{tropo}$ and blowing snow SSA were transported over Hudson

Bay towards the Great Lakes region (Fig. 11a-c). By the time the polar air mass reached the Northeast United States, the BrO $VCD_{tropo}$ were back to normal levels, but because of its longer lifetime, $O_3$ was 5-10 ppbv lower than background mixing ratios as shown by the FYI+MYI Snow simulation (Fig. 11d). This decrease in surface $O_3$ was observed at NAPS and CASTNET surface sites throughout Southeast Canada and the Northeast United States (Fig. 11d).

Ridley et al. (2007) discuss a similar transport event of $O_3$-depleted air from the Canadian Archipelago and nearby Arctic

Ocean to Hudson Bay observed in April 2000 during the Tropospheric Ozone Production about the Spring Equinox (TOPSE) aircraft campaign. During that event, $O_3$ levels were reduced from 30-40 ppbv down to values as low as 0.5 ppbv in a large area below 500 m altitude. Over the 3-year period that we examined, our simulations suggest that transport of $O_3$-depleted air ($\Delta O_3$>20 ppbv) to Hudson Bay occurs 1-10% of the time (0.3-3 days/month) in March and in April (Fig. 9c), with transport of more moderate $O_3$ depletion ($\Delta O_3$>10 ppbv) taking place 10-20% of the time (3-6 days/month). In March-April 2008,

measurements in the ice-free North Atlantic showed transport of $O_3$-poor air from the Arctic basin down to 52° N during the ICEALOT cruise (Gilman et al., 2010): a 13 ppbv decrease in $O_3$ was accompanied by a simultaneous decrease in the acetylene to benzene ratio indicating exposure to halogen oxidation. In our simulation, we find that transport of airmass with $\Delta O_3$>10 ppbv air to sub-Arctic latitudes occurs with a frequency of 5-10% (1.5-3 days/month) down to 50° N and 1-5% (< 1.5 days/month) down to 40° N (Fig. 9d) and that this transport appears to be favored over Hudson Bay extending to the Northeast



United States and the Western Atlantic off Nova Scotia (Fig. 9d). At mid-latitudes (30º-60º N), the FYI+MYI Snow simulation results in a mean surface $O_3$ decrease of 1.2 ppbv (3.3%) in March-April, thus a small but non-negligible contribution.

## 5 Atmospheric deposition on snow as a source of salinity and bromide

We use the results of the FYI Snow simulation to examine the potential role of atmospheric deposition of SSA and gas-phase $Br_y$ as a source of salinity and bromide to snow on sea ice. Snow composition profiles indicate that atmospheric deposition can

sometimes be a more important source of salinity and $Br^-$ than upward brine migration, especially over MYI and in deep snowpack (>10-17 cm) over FYI (Domine et al., 2004; Krnavek et al., 2012; Peterson et al., 2019). Fig. 12a shows the evolution of deposition on sea ice (poleward of 60° N) for $Na^+$ and for total bromine (sum of particulate $Br^-$ and gas-phase $Br_y$) in the GEOS-Chem FYI Snow simulation. Between September and May, deposition of $Na^+$ on snow-covered sea ice has an average daily value of $2.2\times10^6$ kg/day. For total bromine (SSA $Br^-$ and gas-phase $Br_y$), the average daily value is $6.7\times10^4$ kg/day. Open

ocean SSA and blowing snow SSA account nearly equally to these deposition fluxes.

From October to March the deposition of total bromine tracks that of $Na^+$ as both are derived from SSA (Fig. 12a). Starting in late March, deposition of $Na^+$ decreases rapidly following the decrease in blowing snow SSA emissions (Fig. 4c) due to slower wind speeds and warmer temperatures (Huang and Jaeglé, 2017). Deposition of bromine remains high for another month, however, as once bromine activation starts in early spring a large fraction of Br deposition is in the form of $Br_y$, in particular

HBr which is the end product of reactive bromine chemistry. In April-May, deposition of $Br_y$ accounts for more than half of total bromine deposition (Fig. 12a). As $Br_y$ has a longer lifetime against deposition than particulate $Br^-$ and $Na^+$, $Br_y$ shows a more gradual decrease in deposition in April and May. This behavior is consistent with observations of freshly fallen snow at Alert during the Network on Climate and Aerosols Research (NETCARE) campaign, showing a broad spring peak in $Br^-$ deposition between late March and late May (Macdonald et al., 2017). A similar increase of $Br^-$ in snowfall samples after polar

sunrise was reported at Alert by Toom-Sauntry & Barrie (2002). Furthermore, Spolaor et al. (2013, 2014) found a strong seasonal variability of $Br^-$ in polar firn at both Arctic and Antarctic sites, with greater $Br^-$ values in spring/summer compared to winter.

We derive the $Br^-/Na^+$ enrichment factor (EF) of deposition relative to seawater composition (which has a $Br^-/Na^+$ mass ratio of 0.00625 g/g):

$$EF = (\frac{[Br^- + Br_y]}{[Na^+]})_{deposition}/(\frac{[Br^-]}{[Na^+]})_{seawater}$$

EF is calculated for each day and each model grid box and then surface-area weighted over sea-ice to obtain the daily timeseries of pan-Arctic EF shown in Fig. 12b. The model-calculated EF is fairly constant between September and February, with values of ~3-10. Starting in late March, the divergence between bromine and $Na^+$ deposition leads to an increase in EF to a value of





14 by the end of April, followed by a more rapid increase in May up to a value of ~60. In May, the mean model-calculated EF
of deposition is 30. These values are remarkably consistent with the EF measured during the NETCARE campaign at Alert
(Macdonald et al., 2017), as shown in Fig. 12b, with mean observed values of EF~7 between September and February, 15 in
April, and 25 in May. Similarly, Toom-Sauntry & Barrie (2002) observed an increase in EF from low enrichment in the dark
winter months (median 1.5-5) to a large enrichment (median 20-72) after polar sunrise. In our simulation of the blowing snow
SSA, we assumed a constant value of EF=5, which appears to underestimate the EF of snow in May by a factor of 2-3. This
springtime increase in the enrichment of $Br^-$ in snow over sea ice could thus act to propagate bromine explosions and ODEs
into May, and could explain our systematic underestimate in observed $VCD_{tropo}$ in May. Using a one-dimensional model, Piot
& von Glasow (2008) compared modeled and observed deposition on snow at Utqiaġvik, demonstrating that 75% of deposited
bromine may be re-emitted into the gas phase as $Br_2$ or BrCl. They proposed that cycles of deposition and re-emissions of
bromine from snow could thus result in a "leap-frogging" process and explain observations of progressive $Br^-$ enrichment of
snow in coastal Arctic regions with distance inland (Simpson et al., 2005). Our simulations suggest that a similar mechanism
could occur in snow over sea ice via repeated cycles of blowing snow SSA sublimation and deposition of reactive bromine.
These cycles could thus be important in propagating bromine explosions in space (onto coastal snow-covered land) as well as
in time on sea ice (into late spring).

We now use the model-simulated $Na^+$ deposition (FYI Snow simulations) on the sea ice covered Arctic ocean to estimate the
salinity of surface snow due to SSA deposition. In our simulation, wet and dry deposition contribute nearly equally to SSA
deposition. Over the 3-month period between February 1 and April 31, the cumulative SSA deposition on Arctic sea ice is
$10.3 \times 10^8$ kg SSA in the FYI snow simulation. Snow depth on sea ice is not calculated in MERRA-2 and reanalyses tend to
have highly uncertain solid precipitation in polar regions (Lindsay et al., 2014; Uotila et al., 2019), so we rely on the limited
observations available. The Warren et al. (1999) climatology, which is based on 1954-1991 Soviet drifting stations
measurements on Arctic MYI, shows that snow depth increases from 29.7 cm to 34.4 cm during these three months, resulting
in 4.7 cm snow accumulation. This is likely an overestimate of the pan-Arctic snow depth because MYI tends to have thicker
snow and these measurements represent conditions from past decades (Kwok and Cunningham, 2008). Indeed, the 2003-2008
AMSR-E snow depth retrievals over FYI compiled by Zygmuntowska et al. (2014) show much lower snow depths, increasing
from 18 cm in February to 19 cm in in May, thus a 1 cm snow accumulation. These two estimates in snow accumulation result
in pan-Arctic snow accumulations of $10^{14}$ kg to $2.2\ 10^{13}$ kg (sea ice extent is 11 million $km^2$ and fresh snow has a density of
200 $kg/m^3$), which combined with our calculated SSA deposition results in a mean salinity of 0.01 psu - 0.04 psu of surface
snow due to deposition. Fig. 12c shows the spatial distribution of snow salinity due to SSA deposition assuming a 1 cm snow
accumulation. High salinities (>0.05 psu) are calculated in sea-ice regions near the ice-free open ocean in the North Atlantic
and Bering Sea, with lower values on FYI (0.02-0.05 psu) and MYI (0.01-0.03 psu). Deposition could thus represent a
significant source of salinity of surface snow on MYI by the time sunrise reaches high latitudes, and it could also be important
for FYI as the snow gets deeper in late spring (Petty et al., 2018; Warren et al., 1999).



# 6 Conclusions

We used the GEOS-Chem model to examine the impact of blowing snow SSA on bromine chemistry and $O_3$ during Arctic spring. We conduct two blowing snow simulations assuming a 0.1 psu surface snow salinity on FYI and different salinities
on MYI (0.01 psu for the FYI Snow simulation and 0.05 psu for the FYI+MYI Snow simulation). We evaluated these simulations against satellite observations of $VCD_{tropo}$ from GOME-2 and OMI for 2007-2009. We found that our simulations reproduce the spatio-temporal distribution of the observed $VCD_{tropo}$ to within -12% to +17% in March and April and capture the spatial extent as well as the frequency of most large BrO hotspots in March and April. The FYI+MYI Snow simulation captured observed BrO hotspots over the North Pole better than the FYI Snow simulation. In our FYI+MYI simulation,
blowing snow SSA emissions on MYI account for 20-30% of the $VCD_{tropo}$ enhancement and thus represent a significant source of bromine activation. In the past, the search for halogen-containing substrates in polar regions has often focused on the role of frost flowers, newly formed sea ice, and FYI snowpack on polar halogen activation, because of the higher salinity of these surfaces and of their snowpack relative to MYI (e.g., Abbatt et al., 2012; Jacobi et al., 2006; Kaleschke et al., 2004; Simpson et al., 2007a; Yang et al., 2010). Our simulations suggest that the role of MYI in halogen activation is likely also important,
consistent with results of several studies (Choi et al., 2018; Gilman et al., 2010; Halfacre et al., 2014; Peterson et al., 2019).

Inclusion of blowing snow SSA in our FYI+MYI Snow simulation results in a pan-Arctic decrease in surface $O_3$ of 3.7 ppbv (11%) in March and 8.3 ppbv (23%) in April. As most of this decrease is confined to altitudes below 1000 m, the Arctic tropospheric $O_3$ burden decreases by only 3-6% in March-April. Compared to surface $O_3$ observations at coastal Arctic sites, we find that our simulation often underestimates the magnitude of observed ODEs and sometimes misses observed ODEs. The
simulation does reproduce an event of low $O_3$ Arctic air transport down to 40° N, which was observed at multiple sub-Arctic surface sites. We estimate that these transport events occur in March and April with a frequency of 1.5-3 days/month down to 50° N and < 1.5 days/month down to 40° N.

The mixed success of our simulation at capturing the rapid $O_3$ variations observed at coastal Arctic sites could be related to the coarse resolution of the model, the difficulty in simulating Arctic boundary layer exchange processes, the lack of detailed
chlorine chemistry, or the fact that we did not include direct halogen activation by snowpack chemistry. The influence of snowpack bromine chemistry was examined in two previous modeling studies (Falk and Sinnhuber, 2018; Toyota et al., 2011). These studies displayed remarkable agreement with surface $O_3$ observations, in particular at Alert and Utqiaġvik, and reproduced the synoptic variability in GOME VCD. However, both studies systematically underestimated the magnitude of observed BrO VCDs by at least a factor of 2. These results together with our GEOS-Chem blowing snow simulations suggest
that most of the $VCD_{tropo}$ enhancements observed by satellites can be explained by blowing snow SSA leading to BrO enhancements over larger vertical (0-2 km) and horizontal scales (pan-Arctic) than snowpack chemistry, but that in the shallow boundary layer (~50-250 m) over the springtime Arctic direct snowpack halogen activation dominates Br release and is responsible for the most severe ODEs. Indeed, some of the strongest ODEs observed at the surface seem to occur when the



stable Arctic boundary layer is decoupled from convective exchange with the free troposphere (Moore et al., 2014; Seabrook
et al., 2011; Wang et al., 2019). Furthermore as boundary layer $O_3$ reaches very low levels (<4 ppbv), BrO production via R3
is suppressed and atomic Br becomes much more abundant than BrO (Neuman et al., 2010; Wang et al., 2019), such that
satellites would not necessarily observe co-located $O_3$ depletion and enhanced BrO $VCD_{tropo}$.

Based on our analysis of the seasonal variation in modeled deposition of $Na^+$ and bromine on snow, we propose that the
progressive enrichment of bromine in deposition onto sea ice could help propagate blowing snow SSA bromine activation into
May, even as the magnitude of blowing snow SSA emissions starts to decrease. The increase of our calculated springtime
enrichment of Arctic snow $Br^-$ is similar to observations showing a peak after polar sunrise (Macdonald et al., 2017; Spolaor
et al., 2013; Toom-Sauntry and Barrie, 2002). Recycling of deposited bromine was previously proposed in the context of frost
flower/snowpack activation followed by inland transport over coastal regions (Domine et al., 2004; Piot and Glasow, 2008;
Simpson et al., 2005, 2007b). We propose that a similar mechanism takes place over sea ice and snow-covered coastal regions
with blowing snow SSA, enabling BrO explosions to propagate spatially and to last longer into late spring. We also show that
SSA deposition to surface snow in winter-spring, when snow accumulation on sea ice is at its minimum, could account for
0.01-0.03 psu on MYI, 0.02-0.05 psu on FYI, and >0.1 psu on sea ice areas close to the open ocean. While upward migration
of brine from sea ice is the main source of salinity in surface snow over FYI with shallow snowpack, atmospheric deposition
could thus be the dominant source in surface snow over the less saline MYI sea ice and could play an important role in late
spring as the snowpack deepens over FYI (Domine et al., 2004; Krnavek et al., 2012; Peterson et al., 2019).

Our simulations did not include 2-way coupling between snowpack composition and atmospheric deposition, did not consider
blowing snow on coastal snow enriched in $Br^-$, and did not incorporate direct snowpack halogen activation. Incorporating
these coupled processes together with blowing snow SSA promises to yield further insights into the mechanisms leading to
polar bromine activation and ODEs.

**Data availability**. The GOME-2 $VCD_{tropo}$ data is available upon request from Nicolas Theys (Nicolas.Theys@aeronomie.be).
The OMI $VCD_{tropo}$ data is available upon request from Sungyeon Choi (sungyeon.choi@nasa.gov). The surface $O_3$
observations are available from the Observations from these surface monitoring sites are obtained through the World Data
Centre for Reactive Gases (WDCRG, http://ebas.nilu.no), CAPMoN (http://donnees.ec.gc.ca/data/air/monitor/monitoring-of-
atmospheric-gases/ground-level-ozone/),          NAPS          (https://www.canada.ca/en/environment-climate-change/services/air-
pollution.html), and CASTNET (https://www.epa.gov/castnet/castnet-ozone-monitoring) data repositories. The GEOS-Chem
simulations are available upon request from the corresponding author (jaegle@uw.edu).



**Author contributions.** JH and LJ designed the study, conducted the GEOS-Chem simulations, and analysed the results. QC, BA, TS, ME contributed to model development. NT and SC provided the tropospheric BrO columns for GOME-2 and OMI. JH wrote the paper with significant contributions from LJ. All the authors contributed to editing the paper.


**Competing interests.** The authors declare that they have no conflict of interest.

**Acknowledgements.** NASA is acknowledged for its financial support under awards NNX15AE32G and 80NSSC19K1273.

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

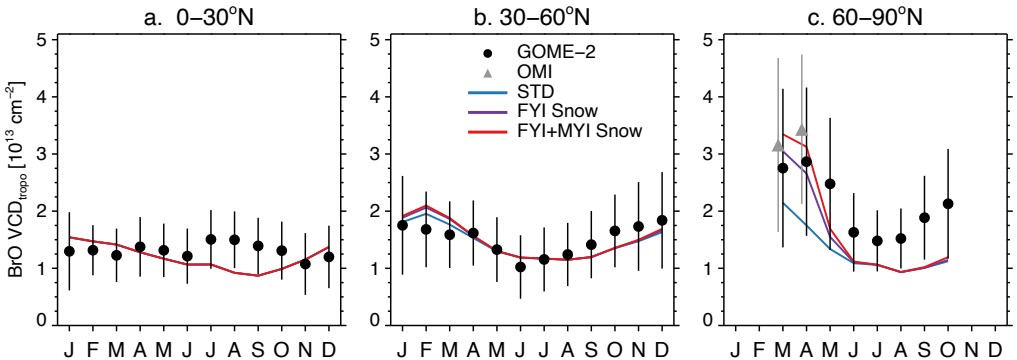

**Figure 1: Seasonal variation in 2007-2009 monthly mean tropospheric BrO vertical column densities (VCD$_{tropo}$ in units of $10^{13}$ molecules cm$^{-2}$) observed by GOME-2 (black circles) and OMI (gray triangles in panel c for April and May), and simulated with GEOS-Chem over (a) 0–30° N, (b) 30–60° N and (c) 60-90° N. The three GEOS-Chem simulations shown are the standard simulation (STD, blue line) and two blowing snow simulations (FYI Snow, purple line and FYI+MYI Snow, red line). The black and gray error bars represent 1 standard deviation about the monthly mean GOME-2 and OMI VCD$_{tropo}$. For each latitude bin, we only show means for months where at least 70% of the surface area has valid satellite observations.**

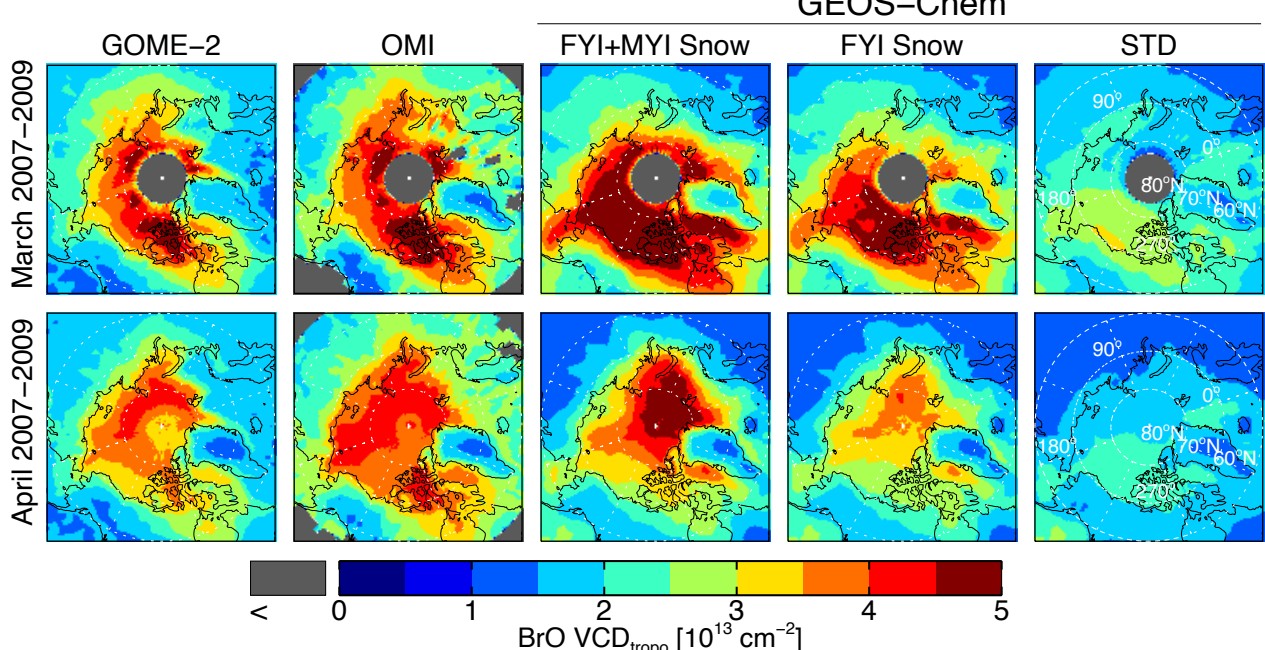

**Figure 2: Spatial distribution of VCD$_{tropo}$ in March (top row) and April (bottom row) 2007–2009. Satellite retrievals from GOME-2 and OMI are compared to the GEOS-Chem FYI+MYI Snow, FYI Snow, and STD simulations.**



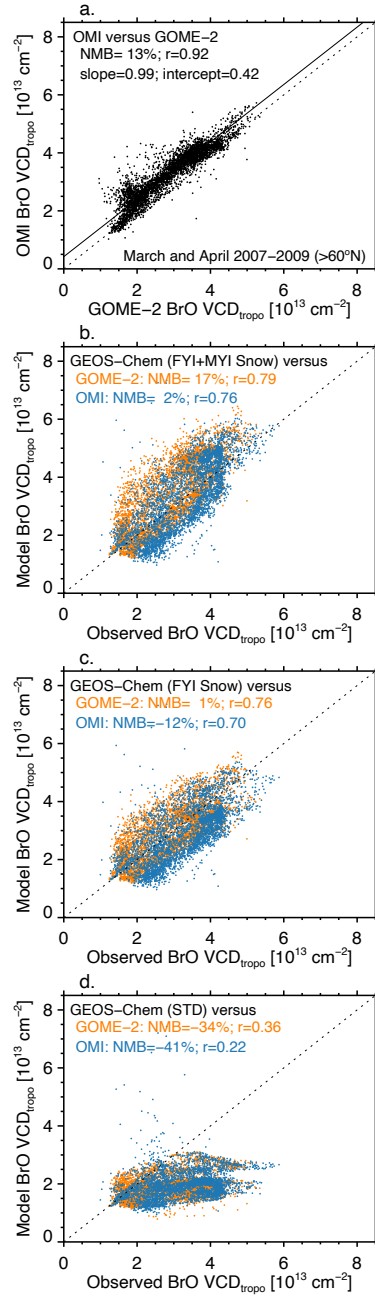


**Figure 3: (a)** Scatterplot of GOME-2 and OMI monthly mean VCD$_{tropo}$ for March and April 2007-2009 poleward of 60° N. **(b)** Scatterplot of VCD$_{tropo}$ calculated with the FYI+MYI Snow simulation and retrieved by GOME-2 (orange circles) and OMI (blue circles). **(c)** Same as (b) but for the FYI Snow simulation. **(d)** Same as (b) but for the STD simulation. The Normalized Mean Bias (NMB, see definition in section 3.1), pearson correlation coefficient, and slope are shown in each panel. For panel b), the slope and intercept of the reduced-major-axis regression line (solid line) are also indicated. The dashed line corresponds to the 1:1 line.


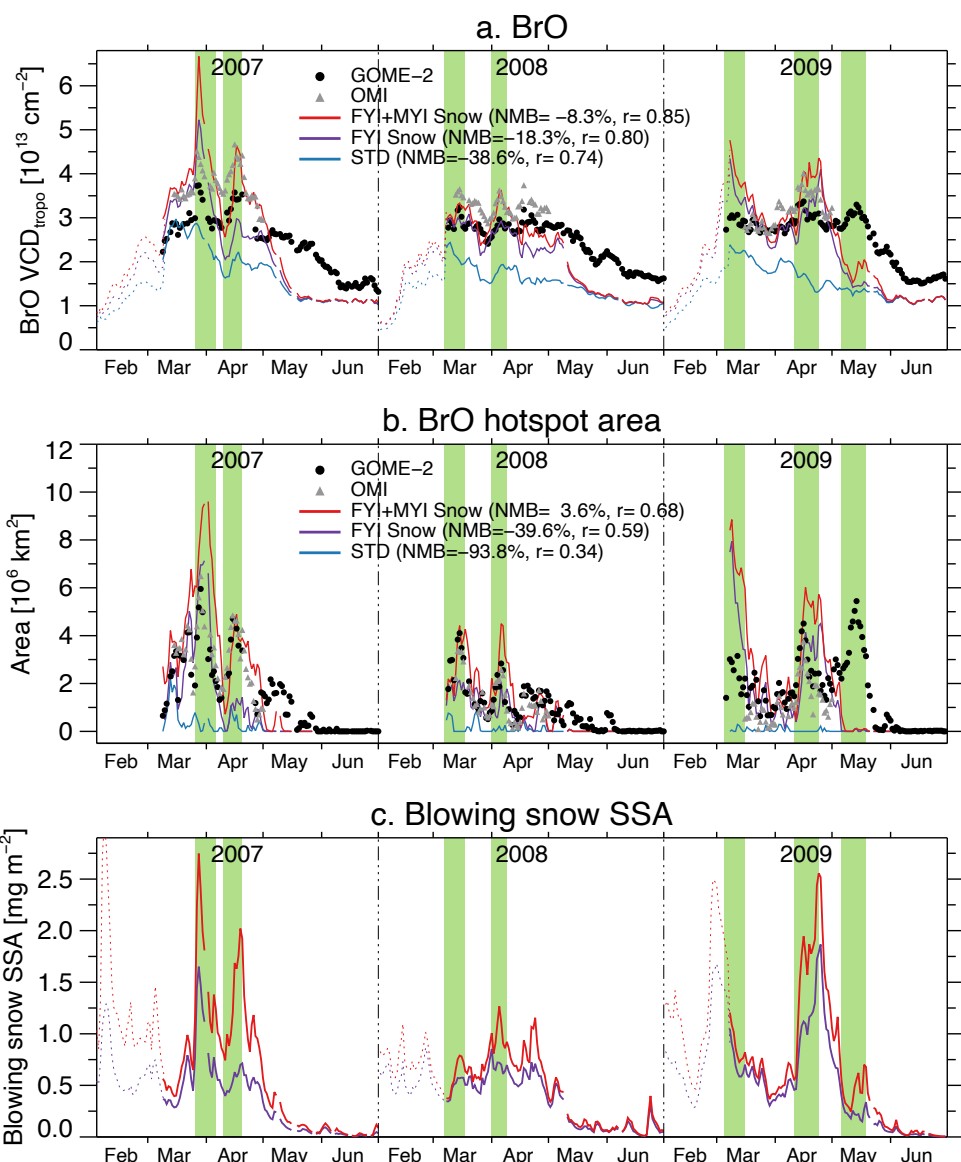

**Figure 4: Timeseries of daily mean BrO VCD$_{tropo}$, tropospheric BrO hotspots area, and blowing snow SSA burden over high latitudes (>60° N) between February and June for 2007−2009. (a) Timeseries of daily VCD$_{tropo}$ (in $10^{13}$ cm$^{-2}$) averaged poleward of 60° N for GOME-2 (black circles), OMI (gray triangles) and simulated with GEOS-Chem (FYI+MYI Snow: red, FYI Snow: purple, STD: blue). (b) The daily area extent of BrO hotspots (in units of $10^6$ km$^2$) for GOME-2, OMI, and GEOS-Chem. See section 3.2 for the definition of BrO hotpot areas. Note that the total area poleward of 60° N is $34.2 \cdot 10^6$ km$^2$. (c) Timeseries of daily mean blowing snow SSA burden (mg m$^{-2}$) for the FYI+MYI Snow and FYI Snow simulations. The blowing snow SSA burden is obtained as the difference between the blowing snow and STD simulations. The normalized mean bias (NMB) and pearson correlation coefficients displayed in panels (a) and (b) are relative to the combined GOME-2 and OMI timeseries. The events highlighted in light green are defined as periods when GOME-2 or OMI BrO hotspots cover more than $2 \cdot 10^6$ km$^2$ for longer than 5 days. We only show days where valid satellite observations are available over at least 70% of the surface area poleward of 60° N. The dashed lines in (a) and (c) correspond to the GEOS-Chem daily means poleward of 60° N for days with less than 70% of valid obsevations.**


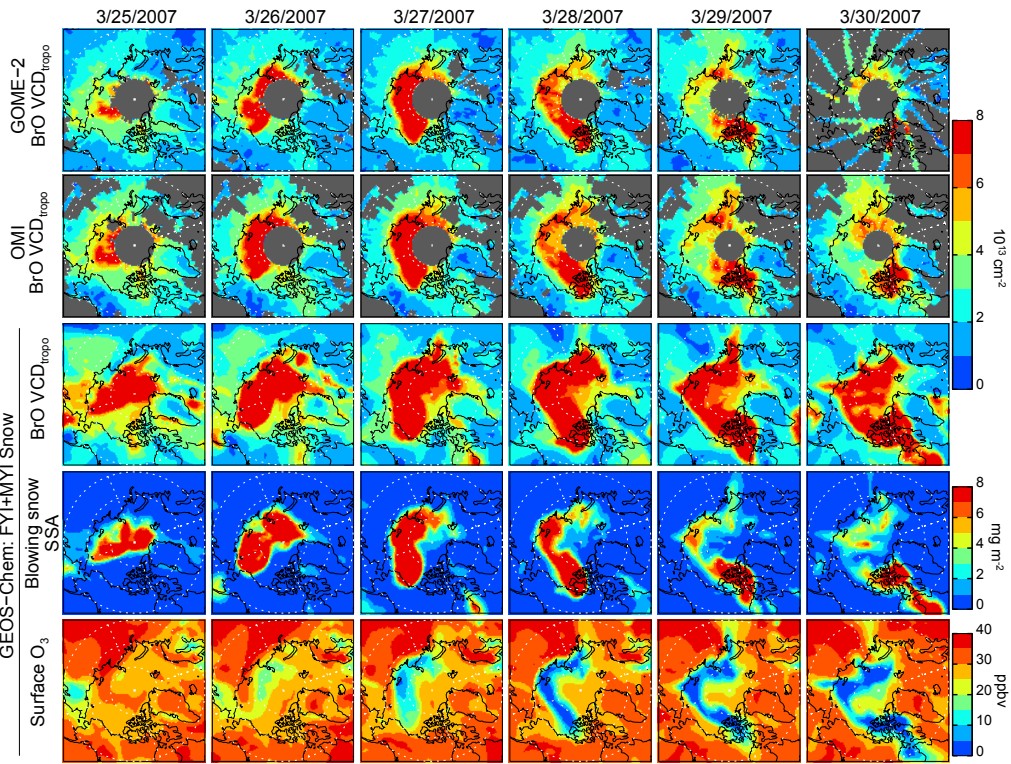

**Figure 5: Daily 25-30 March 2007 distribution of BrO VCD$_{tropo}$ for GOME-2, OMI and GEOS-Chem (FYI+MYI Snow simulation). Also shown are the blowing snow SSA mass burdens (mg m$^{-2}$) and surface O$_3$ mixing ratios (ppbv) for the FYI+MYI**
**Snow simulation. The gray areas in the first two rows correspond to regions with no BrO retrievals. Note that polar sunrise occurred equatorward of 80° N.**



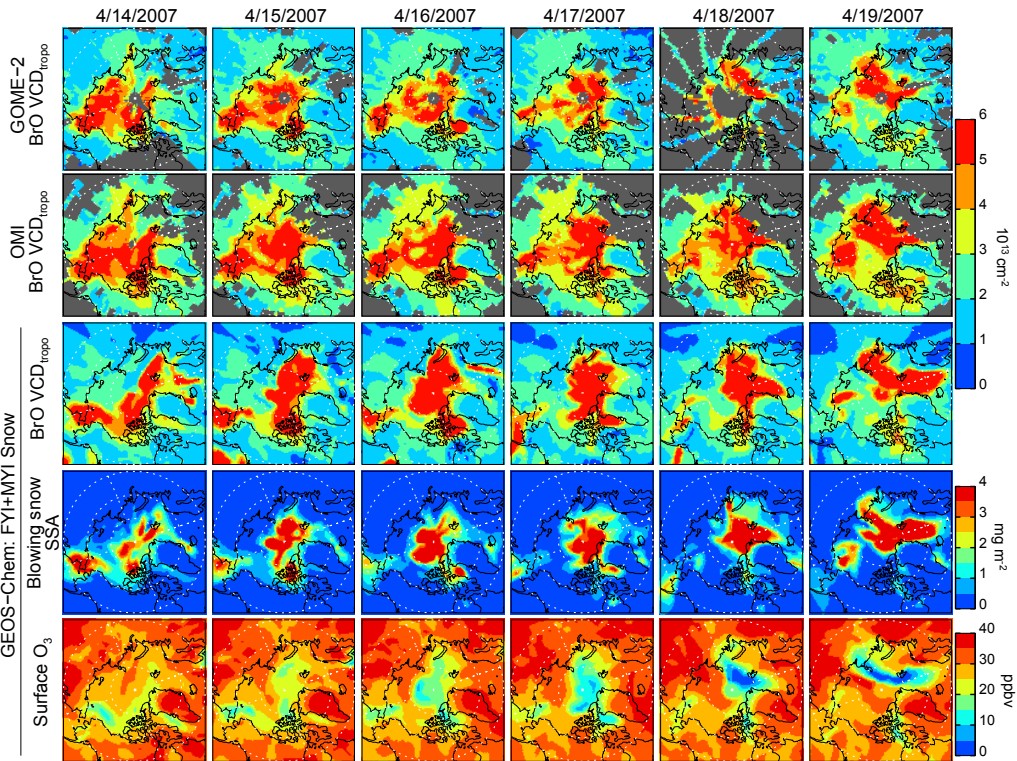

**Figure 6: Daily 14-19 April 2007 distribution of BrO VCD$_{tropo}$ from GOME-2, OMI and GEOS-Chem (FYI+MYI Snow simulation). Also shown are the blowing snow SSA burdens (mg m$^{-2}$) and surface O$_3$ mixing ratios (ppbv) in the FYI+MYI Snow simulation.**

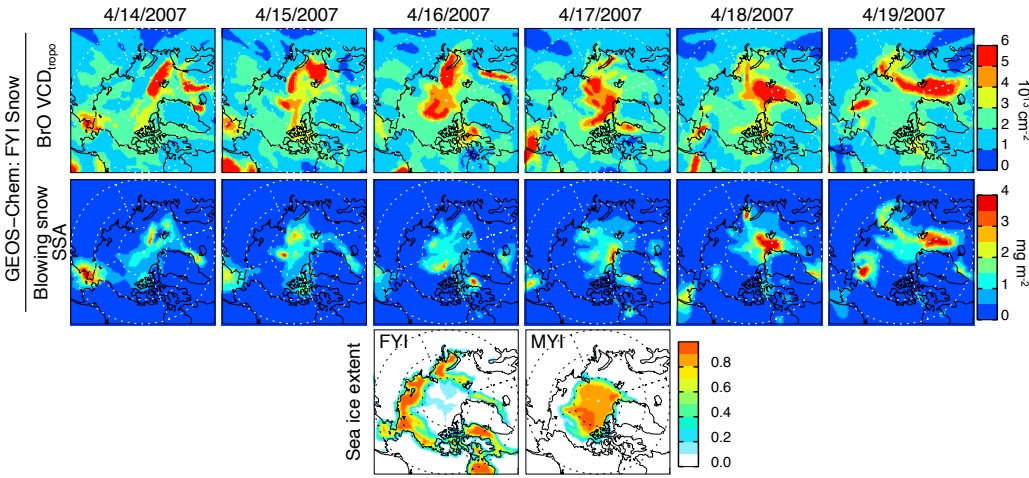

**Figure 7: Daily distribution of BrO VCD$_{tropo}$ and blowing snow SSA burden from the GEOS-Chem FYI Snow simulation for 14-19 April 2007. The bottom row shows the distribution of sea ice extent for FYI and MYI.**


**Figure 8: Timeseries of hourly surface O₃ (ppbv) at three Arctic sites (a,b,c) and 5 sub-arctic sites (d-h) for March 1-May 31 2007: a) Utqiaġvik (Barrow), Alaska, USA, b) Alert, Nunavut, Canada, c) Zeppelin, Spitsbergen, Norway, d) Elk Island, Alberta, Canada, e) Bonner Lake, Ontario, Canada, f) Algoma, Ontario, Canada, g) Egbert, Ontario, Canada h) Woodstock, New Hampshire, USA. In situ observations are shown with the black lines, while the GEOS-Chem STD, FYI Snow, and FYI+MYI**
**Snow simulations are shown in blue, purple, and red, respectively. The shaded green areas corresponding to the times when the large BrO hotspot from 26 March - 4 April 2007 is predicted to be observed at the 3 Arctic sites and then transported to the sub-arctic sites.**

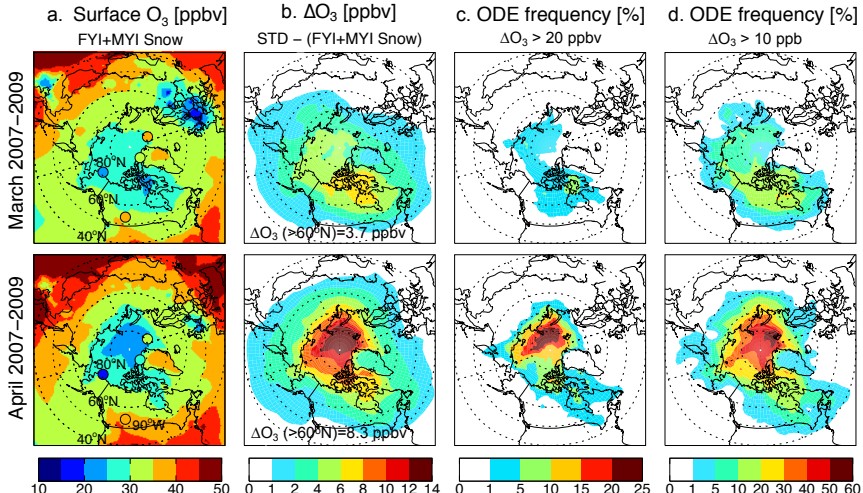


**Figure 9: Monthly mean distribution of surface $O_3$ simulated with GEOS-Chem for March (top row) and April (bottom row) 2007-2009. Column a) Surface $O_3$ mixing ratios in the GEOS-Chem FYI+MYI Snow simulation. The circles are color-coded by monthly mean surface $O_3$ observed at 3 Arctic (Utqiaġvik, Alert, Zeppelin) and 1 sub-Arctic site (Elk Island). Column b) Decrease in surface $O_3$ ($\Delta O_3$, ppbv) due to blowing snow obtained as the difference between the STD and FYI+MYI Snow simulations. Column**
**c): Occurrence frequency of ODEs (in %), defined as the percent of time that more than 20 ppbv of $O_3$ is lost due to blowing snow ($\Delta O_3 > 20$ ppbv). Column d): Occurrence frequency of ODEs, defined as $\Delta O_3 > 10$ ppbv.**

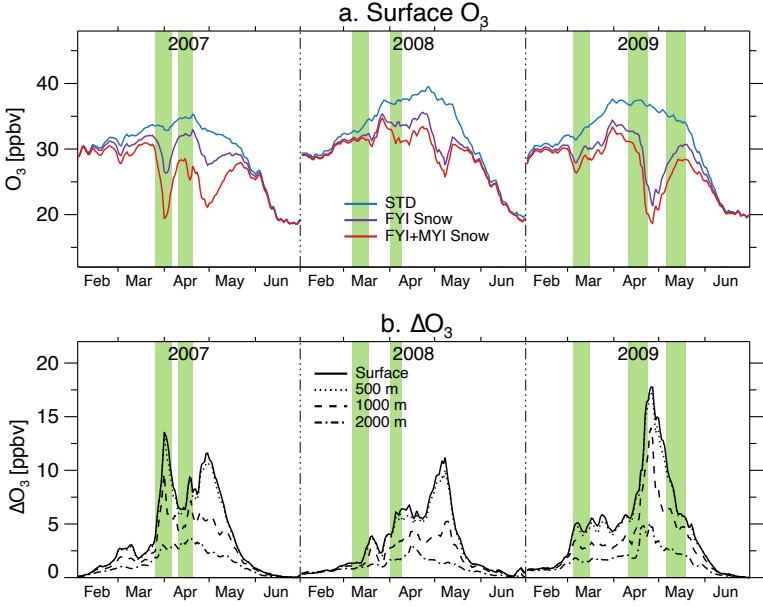

**Figure 10: Timeseries of daily mean surface $O_3$ and decrease in surface $O_3$ ($\Delta O_3$) due to blowing snow at high latitudes (>60°N)**
**between February and June 2007−2009. (a) Daily mean surface $O_3$ mixing ratios averaged poleward of 60° N for the STD (blue line), FYI Snow (purple) and FYI+MYI Snow (red line) GEOS-Chem simulations. (b) Daily mean $\Delta O_3$, obtained as the difference between the FYI+MYI Snow and STD simulations at the surface (solid line), 500 m altitude (dotted line), 1000 m altitude (dashed line), and 2000 m altitude (dashed-dotted line). The events highlighted in light green are defined as periods when GOME-2 or OMI BrO hotspots cover more than $2 \cdot 10^6$ km² for longer than 5 days.**


**Figure 11: Mean 5-9 April 2007 VCD$_{tropo}$ for (a) GOME-2 and (b) the GEOS-Chem FYI+MYI Snow simulation. Mean 5-9 April 2007 (c) blowing snow SSA burden and (d) surface O$_3$ mixing ratios for April 5-9 2007 from the GEOS-Chem FYI+MYI Snow simulation. Surface O$_3$ observations are shown as color-coded symbols in panel (d), with the squares correspond to the sub-Arctic sites from Fig. 8. The letter numbering (e-h) corresponding to the panel numbers in Fig. 8.**

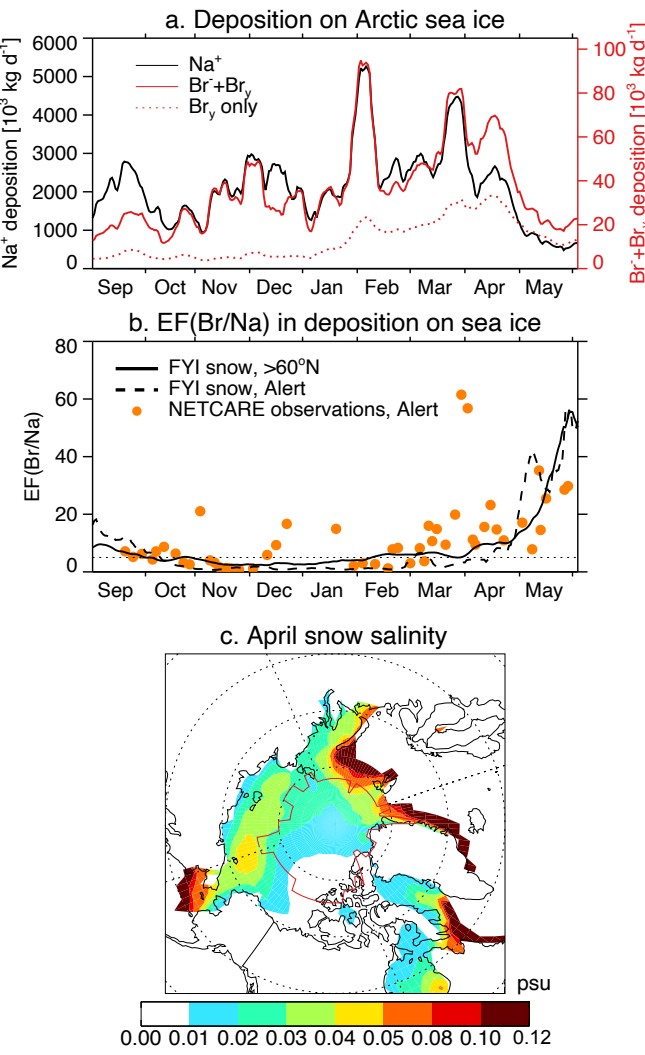

**Figure 12: Modeled evolution of deposition and Br/Na enrichment factor on sea ice (>60° N) for 2007 calculated with the FYI Snow simulation. (a) Total deposition on sea ice for Na$^+$ (black line, left axis), Br$^-$+Br$_y$ (red line, right axis), and Br$_y$ only (dashed red line, right axis) in units of 10$^3$ kg/day. (b) Modeled Br/Na enrichment factor EF (black line) relative to seawater in deposition over sea ice, see section 5 for the definition of EF. The orange circles show the enrichment factors observed during the NETCARE campaign at Alert, Nunavut (Macdonald et al., 2017). The dashed black line is the modeled EF sampled at Alert. A 10-day boxcar smoothing has been applied to all the modeled time series between September and May. The horizontal dotted line represents the value EF=5, assumed for the emitted blowing snow SSA. (c) Surface snow salinity due to the cumulative SSA deposition on sea ice for February 1-April 30 assuming a 1cm snow accumulation rate for that period (see text). The red line shows the location of MYI.**