# Peer review of "Evaluating the impact of blowing snow sea salt aerosol on springtime BrO and O3 in the Arctic"

_Atmospheric Chemistry and Physics, 2019_

## Referee Comment (RC1) · Anonymous Referee #2 · 19 Feb 2020

**General comments:**

Huang and co-authors used their GOES-Chem CTM model to examine the influence of blowing snow SSA as a source of bromine on springtime bromine activation and ozone depletion in the Arctic. Comparing to previous modelling work, this work has some significant progresses, e.g. in parameterising bromine flux from sea ice surface and reproducing enhanced BrO events in spring. Moreover, they pointed out the importance of multiyear sea ice as a source of bromine and recycling of deposited bromine to BrO seasonality. This is a decent modelling work which adds novel information to our understanding of polar bromine chemistry and certainly will benefit following on modelling simulations. The manuscript is well-written, and I only suggest a minor revision.

Specific comments:

Abstract (lines 34-36): I doubt your suggestion of inclusion of snowpack Br activation will definitely fix the model difficulty in reproducing local ODEs. I cannot see how you derive this conclusion, as in your manuscript (lines 366-374 and lines 518-520), you pointed out several potential factors that could cause model failure in reproducing local ODEs, which are (1) coarse resolution, (2) lack of chlorine chemistry, (3) difficulty in modelling boundary layer processes and (4) lack of snowpack emission. The model horizontal resolution used in this study is 2X2.5 degree, which is quite coarse. We know that model cannot capture any sub-grid scale events. If we assume a mean wind speed of 20 km per hour, then any transported event with a lifetime of around or less than ~10 hrs will be smoothed and cannot be reproduced by the model. In which case, it demands a high resolution model simulation. In addition, the poor model ability in reproducing shallow boundary layer, especially in polar regions (which is a common issue in most current models) is another killer. As the air exchange rate at the top of the boundary layer largely determines near surface ozone level. Therefore, if the model cannot reproduce well the dynamical process of the boundary layer, it will not reproduce well near surface ozone. For these reasons, I recommend the authors to discuss model limitations, rather than pointing to other possibilities.

Line 67: 'Four' or 'three classes' ?

Line 333-334: A recent modelling work (by Rhodes et al., 2017) focusing on SSA simulation also showed a similar conclusion that when multiyear sea ice is considered as a source of SSA, modelled SSA has the lowest mean standard deviation across the Arctic sites comparing to other experiments.

Rhodes, R. H., Yang, X., Wolff, E. W., McConnell, J. R., and Frey, M. M.: Sea ice as a source of sea salt aerosol to Greenland ice cores: a model-based study, Atmos. Chem. Phys., 17, 9417-9433, https://doi.org/10.5194/acp-17-9417-2017, 2017.

Lines 526-628, you mentioned that '… in the shallow boundary layer (~50-250m) over the springtime Arctic direct snowpack halogen activation dominates Br release and is responsible for the most severe ODEs', do you have any in situ data to support this statement? In my point of view, the snowpack mechanism (as a dominant source of bromine in ODEs) is still a hypothesis without direct in situ data to support. If ODEs are long-distance transported in association with synoptic systems, why cannot the observed events in the shallow boundary layer be transport-related?

---

## Referee Comment (RC2) · Anonymous Referee #1 · 26 Feb 2020

**Review of "Evaluating the impact of blowing snow sea salt aerosol on springtime BrO and $O_3$ in the Arctic"**

This manuscript presents an evaluation of GEOS-CHEM's ability to reproduce observed springtime ODEs and BrO enhancements using a mechanism that produces sea salt aerosol over both first and multiyear sea ice. The evaluation is conducted via comparisons to tropospheric BrO column measurements from GOME-2 and OMI, as well as in-situ ozone measurements at a variety of Arctic locations. Crucially, rather than assuming a uniform snow salinity distribution based on Antarctic measurements, as prior studies have done, the modeled blowing snow SSA production is informed by actual Arctic snow salinity measurements in a variety of sea ice regions. The authors find that the inclusion of SSA production from blowing snow over multi-year sea ice regions produces better agreement with observations than just first year ice regions, and also postulate that remaining disagreements with observations could be resolved by incorporating snowpack production of molecular halogens into their model. This work represents a meaningful advance compared to prior literature on this topic and should be published after some revisions, which I detail below.

**Major Points**

- The authors discussion of improving the model's ability to reproduce ODEs suggest 4 potential sources of difficulty in line 518, but then the paper's abstract and the rest of the conclusion only focus on the inclusion of a snowpack molecular halogen production mechanism. While I agree that this important mechanism should be included in models, the authors could do a better job explaining how the other three sources mentioned might impact the modeled ODEs and why they chose to focus the bulk of the discussion on this mechanism in particular.

- One other potential explanation for the model doing a better job with BrO than $O_3$ is the evaluation via column measurements rather than concentrations. The ozone loss rate is dependent on the BrO concentration (among other things, [*Thompson et al.*, 2017]), but the measured BrO VCD reflects both concentration and vertical profile of BrO [*Sihler et al.*, 2012]. Thus, the same VCD could have very different implications for ozone depletion near the surface depending on the vertical profile of BrO. This issue wouldn't necessarily be resolved by adding another $Br_2$ source. While satellite-based measurements are a needed tool to evaluate performance over large spatial scales, it would also be good, in the future, to evaluate the model using ground-based BrO concentration measurements as the authors did with ozone.

- Section 4.1: I found this section overly qualitative, particularly when compared to the rest of the paper. This issue also pops up in line 33 of the abstract. I would encourage the authors to come up with a more quantitative description of the fraction of ODEs observed at Arctic sites captured by the model, which would strengthen this section of the paper, and allow the authors to avoid phrases like "captures a few" and "misses some" which seem a bit out of place in a scientific publication.

- Section 4.2: This section would potentially be improved by a discussion of the spatial extent of the modeled ODEs and comparison to prior studies. As an example, *Halfacre et al.* [2014] used buoy-based observations to suggest ODEs can have a spatial extent on the order of 100s of km, is this finding reflected by the model?

**Minor Points**

- The units for vertical column density are molecules per $cm^2$. To my knowledge, the omission of molecules is not an appropriate convention.

- Regarding Fig. 3a comparing the GOME and OMI measurements, is the slope calculation from a typical linear regression or an orthogonal distance regression? Since the measurements have comparable uncertainties, the residuals in both x and y should be minimized when determining the line of best fit.

- Line 331: Define what you mean by high wind speed

- Line 371: Missing reference for Cl acid displacement enhancing ODEs

- Line 477: This propagation of bromine inland has been observed [*Peterson et al.*, 2018].

- Fig. 4,8,10,12 Readibility would be improved though the use of variable line styles as well as color. Figure 8 in particular is not readable.

- Fig. 8 Perhaps move the full timeseries to a supplement and change the x axis to only show Mar 15th through April 15th to encompass the green shaded regions at all sites.

**References**

Halfacre, J. W., et al., Temporal and spatial characteristics of ozone depletion events from measurements in the Arctic, *Atmospheric Chemistry and Physics*, *14*(10), 4875–4894, doi:10.5194/acp-14-4875-2014, 2014.

Peterson, P. K., et al., Springtime Bromine Activation over Coastal and Inland Arctic Snowpacks, *ACS Earth and Space Chemistry*, *2*(10), 1075–1086, doi:10.1021/acsearthspacechem.8b00083, 2018.

Sihler, H., et al., Tropospheric BrO column densities in the Arctic derived from satellite: retrieval and comparison to ground-based measurements, *Atmospheric Measurement Techniques*, *5*(11), 2779–2807, doi:10.5194/amt-5-2779-2012, 2012.

Thompson, C. R., P. B. Shepson, J. Liao, L. G. Huey, C. Cantrell, F. Flocke, and J. Orlando, Bromine atom production and chain propagation during springtime Arctic ozone depletion events in Barrow, Alaska, *Atmospheric Chemistry and Physics*, *17*(5), 3401–3421, doi:10.5194/acp-17-3401-2017, 2017.

---

## Author Comment (AC1) · 17 Apr 2020

Response to reviews on manuscript acp-2019-1094 "Evaluating the impact of blowing snow sea salt aerosol on springtime BrO and O$_3$ in the Arctic"

We thank both reviewers for their insightful and constructive comments. We list our responses in *blue* below.

**Responses to Referee #1**
**Major Points**
• The authors discussion of improving the model's ability to reproduce ODEs suggest 4 potential sources of difficulty in line 518, but then the paper's abstract and the rest of the conclusion only focus on the inclusion of a snowpack molecular halogen production mechanism. While I agree that this important mechanism should be included in models, the authors could do a better job explaining how the other three sources mentioned might impact the modeled ODEs and why they chose to focus the bulk of the discussion on this mechanism in particular.
*We have modified the abstract to include these 4 potential sources of difficulty, instead of just focusing on snowpack chemistry (also see our response to Referee #2). We have also modified the conclusions to include more discussion of these other sources of error.*

• One other potential explanation for the model doing a better job with BrO than O3 is the evaluation via column measurements rather than concentrations. The ozone loss rate is dependent on the BrO concentration (among other things, [Thompson et al., 2017]), but the measured BrO VCD reflects both concentration and vertical profile of BrO [Sihler et al., 2012]. Thus, the same VCD could have very different implications for ozone depletion near the surface depending on the vertical profile of BrO. This issue wouldn't necessarily be resolved by adding another Br2 source. While satellite-based measurements are a needed tool to evaluate performance over large spatial scales, it would also be good, in the future, to evaluate the model using ground-based BrO concentration measurements as the authors did with ozone.
*Yes, indeed, evaluation of this simulation against ground-based BrO is another useful approach and is one that our colleagues at the University of Alaska and University of Florida are examining (Swanson et al., 2019).*

*Swanson, W., Confer, K., Holmes, C.D., Huang, J., Jaeglé, L., and Simpson, W.R., Evaluating GEOS-Chem with snow aerosol source against ground observations of Arctic reactive bromine events, Abstract A51H-2758 presented at 2019 Fall Meeting, AGU, San Francisco, CA, 9-13 Dec.*

• Section 4.1: I found this section overly qualitative, particularly when compared to the rest of the paper. This issue also pops up in line 33 of the abstract. I would encourage the authors to come up with a more quantitative description of the fraction of ODEs observed at Arctic sites captured by the model, which would strengthen this section of the paper, and allow the authors to avoid phrases like "captures a few" and "misses some" which seem a bit out of place in a scientific publication.
*Point well taken. We have revised the comparison between observed and modeled surface O$_3$ to be more quantitative:*
*"Figure 8 shows that FYI+MYI Snow simulation reproduces only 25-30% of the ODEs at*

*Utqiaġvik and Alert, such as the 8 – 18 April and 26 April – 10 May depletion events at Utqiaġvik as well as the April events at Alert. The model predicted magnitude of O₃ depletion at those sites is a factor of two lower than observed. The FYI+MYI Snow simulation performs somewhat better at Zeppelin, where it captures the timing and magnitude of 40% of the observed events, in particular between late March and late April. Our simulation thus misses 60-75% of observed ODEs at those three Arctic sites, in particular several March and late May events at Utqiaġvik, as well as the sustained May events at Alert, and the late May ODEs at Zeppelin."*
*We have also modified the abstract and conclusions accordingly.*

• Section 4.2: This section would potentially be improved by a discussion of the spatial extent of the modeled ODEs and comparison to prior studies. As an example, Halfacre et al. [2014] used buoy-based observations to suggest ODEs can have a spatial extent on the order of 100s of km, is this finding reflected by the model?
*We have added the following sentences to this section to address this:*
*"The median aerial extent of ODEs ($\Delta O_3 > 20$ ppbv) simulated in the FYI+MYI Snow simulation is 0.35 million km² (horizontal extent of ~330 km) with some of the largest ODEs extending over areas of 1.5-7 million km² (~700-1500 km horizontal extent). Our results are consistent with the 282 km median size of major ODEs inferred by Halfacre et al. (2014) using ice-tethered buoys combined with back-trajectories."*

**Minor points**
• The units for vertical column density are molecules per cm2. To my knowledge, the omission of molecules is not an appropriate convention.
*We have added molecules to the units in the text and figures.*

• Regarding Fig. 3a comparing the GOME and OMI measurements, is the slope calculation from a typical linear regression or an orthogonal distance regression? Since the measurements have comparable uncertainties, the residuals in both x and y should be minimized when determining the line of best fit.
*We use the reduced-major-axis regression line to calculate the slope (as noted in the Figure caption, line 912). This is a symmetrical regression method assuming both variables are dependent and have errors. For Fig. 3a, the reduced major axis regression yields the same results as an orthogonal reduced major Axis regression.*

• Line 331: Define what you mean by high wind speed
*>10-12 m s⁻¹. This has been clarified in the revised text.*

• Line 371: Missing reference for Cl acid displacement enhancing ODEs
*We have added a reference to Keene et al. (2007).*

• Line 477: This propagation of bromine inland has been observed [Peterson et al., 2018].
*We have added a reference to Peterson et al. (2018) in the revised manuscript.*

• Fig. 4,8,10,12 Readibility would be improved though the use of variable line styles as well as color. Figure 8 in particular is not readable.

*We have worked to improve the readability of these figures by increasing the line thickness and changing the colors.*

• Fig. 8 Perhaps move the full timeseries to a supplement and change the x axis to only show Mar 15th through April 15th to encompass the green shaded regions at all sites.
*We have chosen to keep the full timeseries for this figure.*

**Responses to Referee #2**

- Abstract (lines 34-36): I doubt your suggestion of inclusion of snowpack Br activation will definitely fix the model difficulty in reproducing local ODEs. I cannot see how you derive this conclusion, as in your manuscript (lines 366-374 and lines 518-520), you pointed out several potential factors that could cause model failure in reproducing local ODEs, which are (1) coarse resolution, (2) lack of chlorine chemistry, (3) difficulty in modelling boundary layer processes and (4) lack of snowpack emission. The model horizontal resolution used in this study is 2X2.5 degree, which is quite coarse. We know that model cannot capture any sub-grid scale events. If we assume a mean wind speed of 20 km per hour, then any transported event with a lifetime of around or less than ~10 hrs will be smoothed and cannot be reproduced by the model. In which case, it demands a high resolution model simulation. In addition, the poor model ability in reproducing shallow boundary layer, especially in polar regions (which is a common issue in most current models) is another killer. As the air exchange rate at the top of the boundary layer largely determines near surface ozone level. Therefore, if the model cannot reproduce well the dynamical process of the boundary layer, it will not reproduce well near surface ozone. For these reasons, I recommend the authors to discuss model limitations, rather than pointing to other possibilities.
*Point well taken. We have modified the abstract to address this and now have a more complete description of other possibilities: "This difficulty in reproducing observed surface ODEs could be related to the coarse horizontal resolution of the model, the known biases in simulating Arctic boundary layer exchange processes, the lack of detailed chlorine chemistry, and/or the fact that we did not include direct halogen activation by snowpack chemistry." We have also expanded the discussion of these other potential sources of error in the conclusions.*

- Line 67: 'Four' or 'three classes'?
*Yes, we have corrected this in the revised manuscript.*

- Line 333-334: A recent modelling work (by Rhodes et al., 2017) focusing on SSA simulation also showed a similar conclusion that when multiyear sea ice is considered as a source of SSA, modelled SSA has the lowest mean standard deviation across the Arctic sites comparing to other experiments.
*We have added a reference to Rhodes et al. (2017) in the revised manuscript.*

- Lines 526-628, you mentioned that '… in the shallow boundary layer (~50-250m) over the springtime Arctic direct snowpack halogen activation dominates Br release and is responsible for the most severe ODEs', do you have any in situ data to support this

statement? In my point of view, the snowpack mechanism (as a dominant source of bromine in ODEs) is still a hypothesis without direct in situ data to support. If ODEs are long-distance transported in association with synoptic systems, why cannot the observed events in the shallow boundary layer be transport-related?

*We recognize that our statement was too strong and have modified it as follows: "… in the shallow boundary layer (~50-250 m) over the springtime Arctic direct snowpack halogen activation could contribute to Br release and potentially be responsible for the most severe ODEs." The support for this statement is in the sentence following the sentence highlighted by the reviewer, in which we cite three studies: "Indeed, some of the strongest ODEs observed at the surface seem to occur when the stable Arctic boundary layer is decoupled from convective exchange with the free troposphere (Moore et al., 2014; Seabrook et al., 2011; Wang et al., 2019a)."*